# Machine learning for microscopy data analytics targeting real-time optical characterization of semiconductor nanocrystals

Amitrajit Mukherjee[1] ✉, Robby Reynaerts [1,9], Bapi Pradhan[1,9], Sudipta Seth[1], Andreas T. Rösch[2], Tamali Banerjee[3], Lata Chouhan[1], Handong Jin[4], Christian Sternemann [5], Michael Paulus[5], Luca Leoncino[6], Kunal S. Mali [1], Steven De Feyter [1], Maarten B. J. Roeffaers [7], E. W. Meijer [2,8], Johan Hofkens [1,8] ✉ & Elke Debroye [1] ✉

Semiconductor nanocrystals with uniform morphology and composition are expected to show consistent responses during light-matter interactions. However, microscopy reveals significant variations in their photoluminescence blinking patterns, even under identical experimental conditions. This discrepancy arises from differences in crystal defects and nonradiative trap states. As a result, heterogeneous blinking patterns serve as valuable indicator of material quality, uncovering several concealed features through statistical analysis of large datasets. Nonetheless, efficient segregation and analysis of numerous blinking trajectories remain a challenge due to laborious calculations, computational bottlenecks, and manual intervention. In this study, we introduce a robust unsupervised machine learning (UML) assisted module to cluster high-dimensional blinking patterns in near-real-time, while calculating category-wise power spectral densities (PSD) to investigate active traps. Furthermore, we explore the impact of data preprocessing on clustering performance. The 'clustering-segregation-analysis' (UML-PSD) methodology demonstrates versatility, paving a way to advance contemporary (micro) spectroscopy, specifically for rapid and cost-effective optical characterization of semiconductor nanocrystals.

Photoluminescence (PL) microscopy is a powerful tool to investigate nanoscale photo-physics of the single molecules[1–7] and semiconductors nanomaterials[8–18] under different experimental conditions. A suitable example is temporally fluctuating PL intensity (or blinking) of conventional quantum dots (QDs)[8–11] and metal halide perovskites (MHP) nano- and microcrystals (NCs and MCs)[12–18]. Blinking signifies involvement of crystal defect-states (traps) in the emission process, which transiently confine photogenerated or injected charge carriers

[1]Division of Molecular Imaging and Photonics, Department of Chemistry, KU Leuven, Leuven, Belgium. [2]Institute for Complex Molecular Systems, Laboratory of Macromolecular and Organic Chemistry, Eindhoven University of Technology, Eindhoven, The Netherlands. [3]Department of Computer Science, Indian Institute of Technology Bombay, Mumbai, India. [4]School of Science, Shandong Jiaotong University, Jinan, China. [5]Fakultät Physik/DELTA, Technische Universität Dortmund, Dortmund, Germany. [6]Electron Microscopy Facility, Istituto Italiano di Tecnologia, Genova, Italy. [7]cMACS, Department of Microbial and Molecular Systems, KU Leuven, Leuven, Belgium. [8]Max Planck Institute for Polymer Research, Mainz, Germany. [9]These authors contributed equally: Robby Reynaerts, Bapi Pradhan. ✉e-mail: amitrajit.mukherjee@kuleuven.be; johan.hofkens@kuleuven.be; elke.debroye@kuleuven.be

(electrons/holes) and steer them to nonradiative (NR) recombination pathways[19]. This hampers intra- and inter-band carrier mobility, reducing the quantum yield and energy conversion efficiency of devices[19–22]. Therefore, understanding the nature of NR traps and temporal variation of their density is crucial to qualitatively evaluate these nanomaterials[23], especially for optoelectronic applications[24,25], which can be achieved by analysing PL blinking characteristics through optical microscopy techniques[26].

Blinking behavior and trap-mediated charge carrier dynamics are usually influenced by external factors, like pulse energy and repetition frequency of the laser[27], temperature[28], size[29] and shape[30] of the NCs, excitation power[31–33] and wavelength[34,35], surrounding atmosphere[14,15], and surface ligands[16,36,37]. Nonetheless, individual NCs, despite a similar composition and morphology, often exhibit diverse blinking patterns under identical experimental conditions, due to inhomogeneous nature of traps (defects) therein[15,16,38]. For instance, blinking propensity involving high (ON) and low (OFF) intensity levels, which represents the radiative and NR recombination (respectively) of charge carriers, can be affected by the deep and shallow nature of traps. Deep traps primarily extend the duration of the OFF interval, while shallow traps generate long ON times[12,15,38,39]. In this way, different types of traps in the NCs lead to miscellaneous and non-ergodic two-state blinking (TSB) dynamics[38]. Intriguingly, MHP-NCs exhibit a variety of blinking patterns, alongside ON-OFF-ON scenario, where PL gradient(s) are evident due to temporally fluctuating trap density[14–16]. Gradual photo-enhancement and blinking (PEB) is observed in presence of atmospheric constituents like $O_2/H_2O$ and light[15,40], or anion-based post-treatments[16]. This implies coordination of electron-rich species to the surface defects that annihilates (part of) the NR traps. Conversely, ambient conditions also degrade the MHP-NCs, leading to temporal increase of trap density, and a photobleaching with blinking (PBB) signature[14–16]. Even blinking involving multiple intensity states (MSB) is reported in presence of transient traps with different NR efficiencies[15,16,32,41]. Therefore, a consensus has emerged that photo-excitation of QDs and NCs yields diverse blinking patterns that can be a measure of their crystal quality.

Traditionally, TSB trajectories are characterized using a threshold-dependent analysis[10,11,16,37,38] which is limited to a particular set of categories, excluding analysis of other crystals with different PL blinking behaviors. This may prejudice the interpretation of stability and optoelectronics of the MHP-NCs. Such limitations can be addressed by threshold-independent power spectral density (PSD) analysis, within the framework of a 'multiple recombination center' (MRC) model of blinking[42–45]. Nonetheless, PSD analysis is yet to be explored for different blinking patterns. Therefore, statistics over numerous heterogeneous blinking traces and PSD analysis of the subpopulations, can uncover various features of carrier dynamics under specific experimental conditions. Thereby, this strategy reduces ambiguity in inferring the optical quality of NCs, especially when it is established that individual blinking pattern implies a specific nature and dynamic density of NR defects. Investigation of statistics as a function of experimental parameters is particularly insightful, as recent studies reported an intriguing change in subpopulation blinking behavior of MHP-NCs upon alteration of atmosphere ($N_2$, Ar, $O_2$, $H_2O$)[15] or chemical post-treatment[16]. However, this technique is rarely employed due to the lack of a programmed interface capable of quickly segregating and analyzing high-dimensional PL blinking trajectories, a task that can be effectively accomplished using machine learning tools.

The present study inaugurates an unsupervised machine learning (UML, K-means clustering[46–54]) module coupled with PSD analysis (UML-PSD), designed to inspect complex charge carrier dynamics of MHP NCs by clustering and analyzing their PL blinking patterns. Additionally, this technique is utilized to identify synchronously blinking grains within an assembly of MCs, enabling quantification of (photogenerated) carrier diffusion process[16–18,55–57]. For this purpose,

synchronously blinking pixels are clustered to generate a cluster map of the assembly, while pixel-wise PSD analysis of blinking trajectories produces a map of relevant parameters, revealing spatially varied nature of defects (traps). Moreover, versatility of the clustering-segregation strategy is tested on scanning tunneling spectroscopy (STS, Supplementary Note 9) data interpretation for 2D self-assembled molecular networks (SAMNs)[58], which efficiently segregates relevant current-voltage (I-V) signals from experimental biases arising at solid-liquid interfaces at room temperature (RT). We discuss the methods, multipurpose usages of our UML-PSD workflow and the supporting results in the Supplementary Notes (1-11), Supplementary Figs. (1–40) and Supplementary Tables (1–3). The Supplementary Fig. 1 to Supplementary Fig. 8 belong to the Supplementary Notes, demonstrating the methods therein. In summary, the state-of-the-art UML-PSD method offers a generalized solution to achieve fast analysis of different (micro)spectroscopy-generated data. With the potential to integrate the entire technique into a single MATLAB code, it could enhance conventional microscopy analyses, providing immediate insights into material properties and advancing research across multiple scientific domains.

## Results
### Clustering protocol for blinking trajectories

The UML-PSD method is primarily applied to segregate and analyse wide-field microscopy generated PL blinking profiles. A miscellaneous test dataset is constructed with major blinking characteristics (PEB, TSB, PBB) comprising 10,000 datapoints (dimensionality), acquired from randomly chosen 204 diffraction-limited spots of cesium lead bromide (CsPbBr$_3$) NCs (Fig. 1a–b), considering various region of interests (ROI). The time resolution of the data acquisition has been set at 30 ms for each ROI to extract blinking data of total 300 s. We consider photobleaching and photo-brightening trajectories to originate from single nanocrystals, as it is highly unlikely that multiple nanocrystals within the same diffraction-limited spot would undergo bleaching or brightening with identical rates. The same applies to two-state blinking, which is widely recognized as the fingerprint of a single nanocrystal or molecule. Based on our expertise in charge carrier dynamics and blinking analysis of NCs (and MCs)[59,60], we are confident that our methodology predominantly reflects single-nanocrystal blinking phenomena. The synthesis procedure of these dodecahedron NCs, the material characterization, the preparation of thin-films, and the data acquisition techniques are discussed in the Methods section. The transmission electron microscopy (TEM) data, the size distribution analysis, the X-ray diffraction (XRD) patterns, and the optical characterization of CsPbBr$_3$ NCs through bulk absorbance, the PL spectroscopy, and the time-averaged wide-field image are presented in the Supplementary Fig. 9a–e.

The segregation of unlabeled blinking patterns (Fig. 1c) sequentially follows three steps, (i) estimation of the optimum cluster ($K_{opt}$), (ii) classification of data, and (iii) cluster-wise PSD analysis of the labeled PL traces. For first two objectives, we deploy unsupervised K-means algorithm[49–54] (Fig. 1d, Supplementary Note 1), utilizing a customized MATLAB (R2022a) code with inbuilt functions. K-means clustering relies on minimum (point-to-point) Euclidean distance (Supplementary Note 2) and is deliberately chosen as spectroscopy always generates pixel-wise blinking trajectories comprising consistent length of information without missing data. To estimate the $K_{opt}$ and cluster the blinking patterns, we have constructed the Euclidean distance matrix based on normalized PL trajectories. By scaling all blinking traces to a fixed range (0 to 1), normalization enables shape-based comparisons which would not be achievable using raw intensity data. Next, information of the $K_{opt}$ and the normalized blinking signals are fed into the K-means module and individual clusters are segregated. During K-means process, blinking traces with 10,000 observations are envisioned as a single point of 10,000 (N) dimensionality

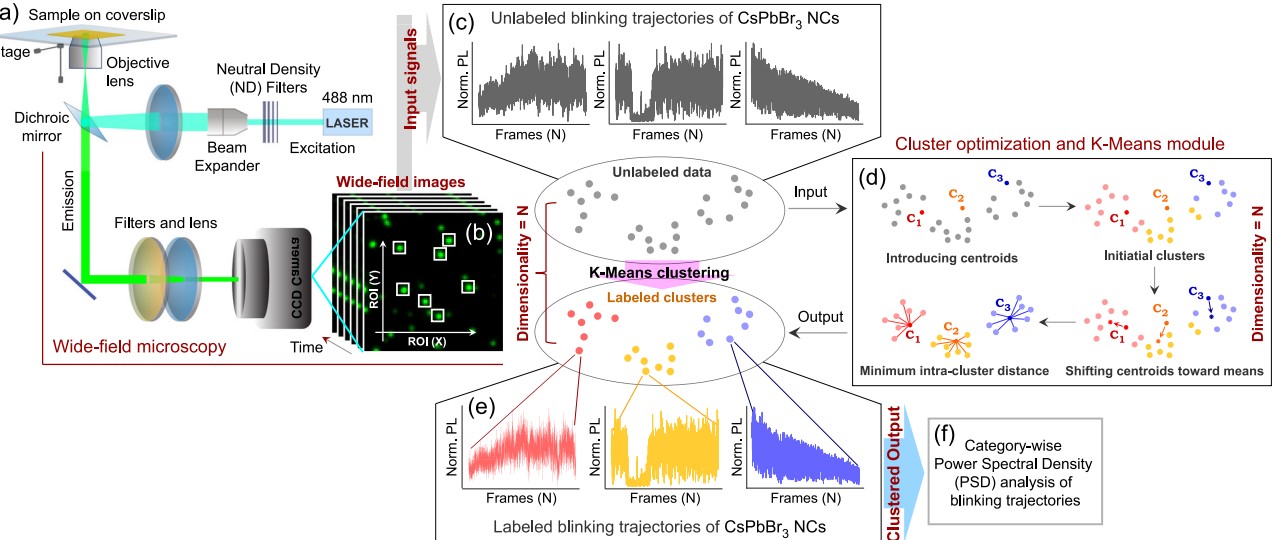

**Fig. 1 | Schematic of the clustering strategy envisioned for PL blinking analyses.** **a** Schematic illustration of wide-field epifluorescence microscopy, and (**b**) time-averaged region of interest (ROI) from a wide-field PL movie (.tiff file) of a CsPbBr$_3$ NCs thin-film, acquired by EM-CCD (electron multiplying charge-coupled device) camera. **c** MATLAB-based automated module that extracts PL blinking data from spatially isolated diffraction-limited spots on the ROI, carefully chosen utilizing the ImageJ software. **d** Illustration of the K-means clustering process, where the accumulated PL trajectories are clustered in 'N' dimensional space, based on minimum Euclidian distance. **e** The labeled datasets of the PL blinking trajectories enabling comprehensive subpopulation statistics, followed by (**f**) cluster-wise power spectral density (PSD) analyses of the blinking data.

(Fig. 1d, Supplementary Fig. 10). The labeled trajectories (Fig. 1e) undergo PSD analysis (Supplementary Note 3, Fig. 1f), where distributions of PSDs are fitted with a power-law function and corresponding exponents (β) are extracted. Subsequently, cluster-wise β values are compared to infer different kinetics of PL fluctuations[42–45]. A schematic representation of UML-PSD methodology for PL microscopy and blinking analysis is depicted in Fig. 1.

## The optimum number of clusters ($K_{opt}$)
For this trial dataset, we execute the 'Visual & Logical' (V&L) module (see Methods) to estimate the $K_{opt}$ value. It calculates statistical metrics, like mean Silhouette score (<SS>)[61–64], mean Calinski-Harabasz index (<CH index>)[65–67], and mean sum square deviation/error (<SSD>, with respect to the centroid)[68–70], performing 100 iterations (epochs) of the code (Supplementary Notes 4–6), for trial clusters $N = 2$ to 15. We find that <SS> and <SSD> profiles consistently decrease as the number of trial clusters increases, with an elbow break emerging at cluster 3 (Supplementary Fig. 11a,c,d,f). This characteristic of the <SS> profile likely originates from the temporal similarities between some of the PEB and TSB blinking trajectories (Supplementary Note 4, Supplementary Fig. 12). The elbow break at cluster 3 reflects the $K_{opt}$ accurately indicating existence of three blinking categories in the ensemble, while <CH index> continuously declines (Supplementary Fig. 11b,e) for increasing value of N with no elbow evident.

## Segregation of PL trajectories
Based on the <SS> and <SSD> metric profiles, the K-means algorithm is initiated for the normalized PL blinking dataset considering $K_{opt} = 3$. However, we observe an inefficient clustering with 34% misclassification (Supplementary Table 1). The misclassification is prominent for the PEB and TSB categories (Supplementary Fig. 13, cluster 2-3), implying their difficult segregation due to partially analogous PL progression of some trajectories. To quantify this resemblance, we incorporate another metric into the V&L module which measures the percentage of misclassification (%Mscl) by segregating the blinking traces that exhibit a negative SS value (SS < 0) within the ensemble. The SS < 0 signifies that the data has been misclassified into inappropriate clusters[61]. The mean %Mscl (<%Mscl>), averaged over 100

iterations, gradually rises (Supplementary Fig. 14) with a minimum value at cluster 2, instead of cluster 3 as anticipated by the <SS> profile. The discrepancy implies that some PEB and TSB traces are apparently similar, situating in the intersection between the datasets in 10,000-dimensional space (Supplementary Fig. 15). These points likely exhibit a reduced SS value, as increase in SS implies elongation of the Euclidean distance between the elements of two adjacent (nearest neighbor) clusters (Supplementary Note 4).

Further, we calculate the percentage of NC emitters (%EM) at different SS levels to evaluate the extent of overlap between blinking datasets. Average %EM (< %EM >) profiles are constructed for SS < 0 up to SS < 0.6, under 100 iterations, for trial clusters $N = 2$ to 15 (Supplementary Fig. 16). We find a sharp change in the slope of <%EM> profiles at cluster 3, considering SS < 0.3 and onwards, which is absent for the profiles corresponding to SS < 0.2 and SS < 0.1. Therefore, the span of intersection may roughly be estimated between 0.2 and 0.3 in the Silhouette space, and elements outside the intersection (SS > 0.3) are likely well-segregated among three blinking categories. At this point, although we refrain from concluding an exact extent of intersection, a sharp change in the slope of <%EM> profiles can be an indicator providing scope to automate the decision in the future. Altogether, our observation indicates that the <%Mscl> and K-means clustering are often influenced by similarities between high-dimensional PL trajectories. The V&L module took approximately 550 s, 970 s, and 190 s to perform the calculations related to SS, CH indexes, and SSD profiles (respectively), for the raw blinking dataset, considering 14 trial clusters and 100 iterations of the codes.

## Effect of data processing on clustering efficiency
To reduce the 'curse of dimensionality' for long-time PL blinking data and the estimation time of the $K_{opt}$, we employ preprocessing (smoothing) of the blinking traces. We perform noise-flattening using discrete wavelet transformation (DWT)[71–74] (Supplementary Note 7) and simple data-binning[75,76] techniques. While DWT-based smoothing retains the actual length of blinking profiles, the binning method reduces both dimensionality and randomness of the data. An optimum data-processing is estimated (*vide infra*) without sacrificing essential blinking features, and thereby the underlying trend component is

accentuated. The processed PL trajectories are then normalized to compare the patterns, creating a secondary input dataset. Next, the $K_{opt}$ is estimated for this secondary dataset, and subsequently K-means clustering is executed. The resulting cluster-indexes guide the retrieval and segregation of the original blinking trajectories, followed by cluster-wise PSD analysis of the raw data.

As technically the DWT at wavelet-level 1 (WL1) returns the pristine signals, we use WL3 to WL7 to smooth the blinking data and recalculate the <SS>, <CH index> and <SSD> metric profiles. Three representative PL intermittency trajectories (PEB, TSB, PBB), processed at different WLs are shown in Supplementary Fig. 17. The V&L module demonstrates that the <SS> profiles decline continuously (for all WLs) with an elbow break at cluster 3 (Supplementary Fig. 18a), representing the $K_{opt}$, which disappears for WL7 possibly due to over-smoothing of the trajectories. The <SS> values gradually increase for WL3 to WL6, as compared to the raw data, indicating improved distinguishability of the clusters. The maximum <SS> at the elbow is achieved for WL6, which simultaneously produce a minimum <%Mscl> (Supplementary Fig. 18b) for WL6 (0.32%, out of 100 iterations). It is important to note here that the minimum misclassification at cluster 3 is evident specifically for the WL-smoothed traces, indicating importance of pre-processing on their distinguishability in the ensemble. The <CH index> profiles further exhibit a prominent elbow (Supplementary Fig. 18c) at cluster 3, which is however not evident in case of the unprocessed dataset. Once again, the maximum value of the <CH index> at the elbow is observed for WL6. This is in tune with the normalized <SSD> profiles which yield the elbow at cluster 3, being minimum for WL6 (Supplementary Fig. 18d). Hence, the V&L module suggests $K_{opt} = 3$ and WL6 to be the optimum scale of data smoothing. Next, the WL6-smoothed blinking traces are subjected to the K-means clustering module and a significant improvement in clustering efficiency (Supplementary Fig. 19) is noted with an accuracy of 88.7% (Supplementary Table 2). The analysis times for SS related calculations, CH index, and SSD profiles reduce to approximately 460 s, 580 s, and 157 s (respectively), for 100 iterations and 14 trial clusters.

To further decrease the estimation time of $K_{opt}$, which is crucial to segregate the microscopy signals in real-time, we reduce the dimensionality of PL traces by data-binning without compromising the essential blinking signatures. Binning windows of W = 10, 20, 25, 40, 50, 80, 100, 125, 200, 250, 400 and 500 frames (equivalent to W×30 ms time-bins) are applied to the original blinking profiles and the ideal scenario is estimated relying on the <SS>, <CH index> and <SSD> metric profiles. Three representative PL trajectories, binned by different frame windows, are shown in the Supplementary Fig. 20. For each binned dataset, the calculated SS values are much higher compared to the pristine blinking signals. The <SS> profiles decline with increasing number of trial clusters after an elbow at cluster 3 (Fig. 2a). It is intriguing that the blinking data binned by W = 40, 50, 80, and 100 frames, display maximum <SS> value at the elbow point (Fig. 2a), which is highest for the binning window of 80 frames. This condition produces least <%Mscl> (0.044 out of 100 iterations) at cluster 3 (Fig. 2b), which is even lower compared to smoothing by WL6 (Supplementary Fig. 18b). Again, the minimum misclassification at cluster 3 is achieved for the binned trajectories, instead of raw traces, which directly imply relevance of data-smoothing on the distinguishability of patterns in the ensemble. In agreement, <CH index> profiles comprise the elbow with a maximum value (Fig. 2c) at cluster 3 for the datasets binned by 25, 40, 50, 80, 100 frames, which is certainly not present in case of indigenous PL blinking trajectories. The <CH index> value at the elbow is highest for the binning window of 80 frame. The elbow also appears in the declining <SSD> profiles (normalized) at cluster 3 (Fig. 2d), with a minimum value evident for the binning window of 80 frames. Unnormalized <SSD> profiles are demonstrated in Supplementary Fig. 21. Thereby, the metrices estimate the $K_{opt} = 3$ and optimum binning window to be 80 frames. The representative blinking

traces processed by WL6 and a bin window of 80 frames are shown in the Supplementary Fig. 22. The V&L module records the analysis time to be approximately 16 s, 28 s, and 8 s for the SS, CH index and SSD related calculations, considering 100 iterations and 14 trial clusters. A significant 20-to-30-fold reduction in the estimation time of $K_{opt}$ as compared to the pristine and DWT (WL6) processed PL trajectories is particularly intriguing. Next, the binned (by 80 frames, 2.4 s) blinking data is introduced to the K-means module, which shows an increase of clustering efficiency (Supplementary Fig. 23) with 89.2% accuracy (Supplementary Table 3). Altogether, our results establish the advantage of data preprocessing, particularly the (time) binning approach, to achieve efficient and rapid clustering of blinking trajectories. We emphasize here that the normalization of blinking patterns is an essential step for shape-dependent clustering. Accurate clustering of unnormalized raw blinking data is challenging, as the wide variation in intensity values leads to inevitable misclassifications even after standard data pre-processing (Supplementary Figs. 24–25).

## Cluster-wise power spectral density analysis

After clustering the binned PL trajectories, the algorithm retrieves original (raw) blinking traces (10,000 dimensionality, of 300 s) based on their cluster index and categorically performs PSD analysis. For each cluster of trajectories (Supplementary Fig. 26a–c), we find that the distribution of PSDs exhibits a power-law dependence over three orders of magnitude in both the frequency and PSD domains. The power-law dependency of the PSDs can be explained within the framework of the MRC model[42,43], suggesting that the rate of switching cycle between active and passive modes of NR traps spans at least three orders of magnitude. This is likely because different types of NR traps are transiently operational in individual nanocrystals.

The exponent (β) of the power-law characteristic essentially relates to the information regarding the frequency of PL intensity oscillation and their contribution to the blinking trajectory, which can be attributed to the crystal defects and nature of the traps[44,45]. For instance, blinking traces incorporating major contributions from the high-amplitude intensity oscillations increase the β value, presumably due to long-lived deep NR traps associated with the MRCs. Besides, PL fluctuations resembling a 'shot noise' characteristic can significantly reduce the β value, which can be referred to the fast switching between short-lived, presumably shallow trap states[45]. The complete diagram of the UML-PSD workflow is illustrated in Fig. 3a–d. A considerable number of PEB and TSB traces exhibit an exponent value greater than 1 and their means (<β>) are equivalent (Supplementary Fig. 26d), which indicates similarity in their PL fluctuations (Supplementary Fig. 12). This aligns with the conjecture of intersecting PEB and TSB datasets, as derived from SS-based calculations. Active deep traps can be anticipated for both cases, which is mostly evident in PEB trajectories after prolonged photoexcitation. A considerable number of PBB trajectories show β values much less than 1, implying fast fluctuation of MRCs, likely associated with shallow traps, below the temporal resolution (30 ms) of blinking data acquisition.

## Challenges with multi-state blinking trajectories

Besides typical blinking signatures, multi-state blinking profiles (MSB, Supplementary Fig. 9e) are often evident for nanocrystals, even though we carefully discarded the clusters (or aggregates) from the analysis (see Methods, Data extraction). Importantly, it is difficult to understand whether MSB characteristics originate from a single or multiple nanocrystals present within the diffraction limit. This warrants additional experiments like combined Fluorescence and Scanning Electron Microscopy (Fluo-SEM), antibunching experiments etc., which are time consuming to perform for a large ensemble of nanocrystals. This necessitates segregation of MSB trajectories as these are complex to analyze and interpret in terms of charge carrier dynamics. We find clustering of the blinking features become challenging when

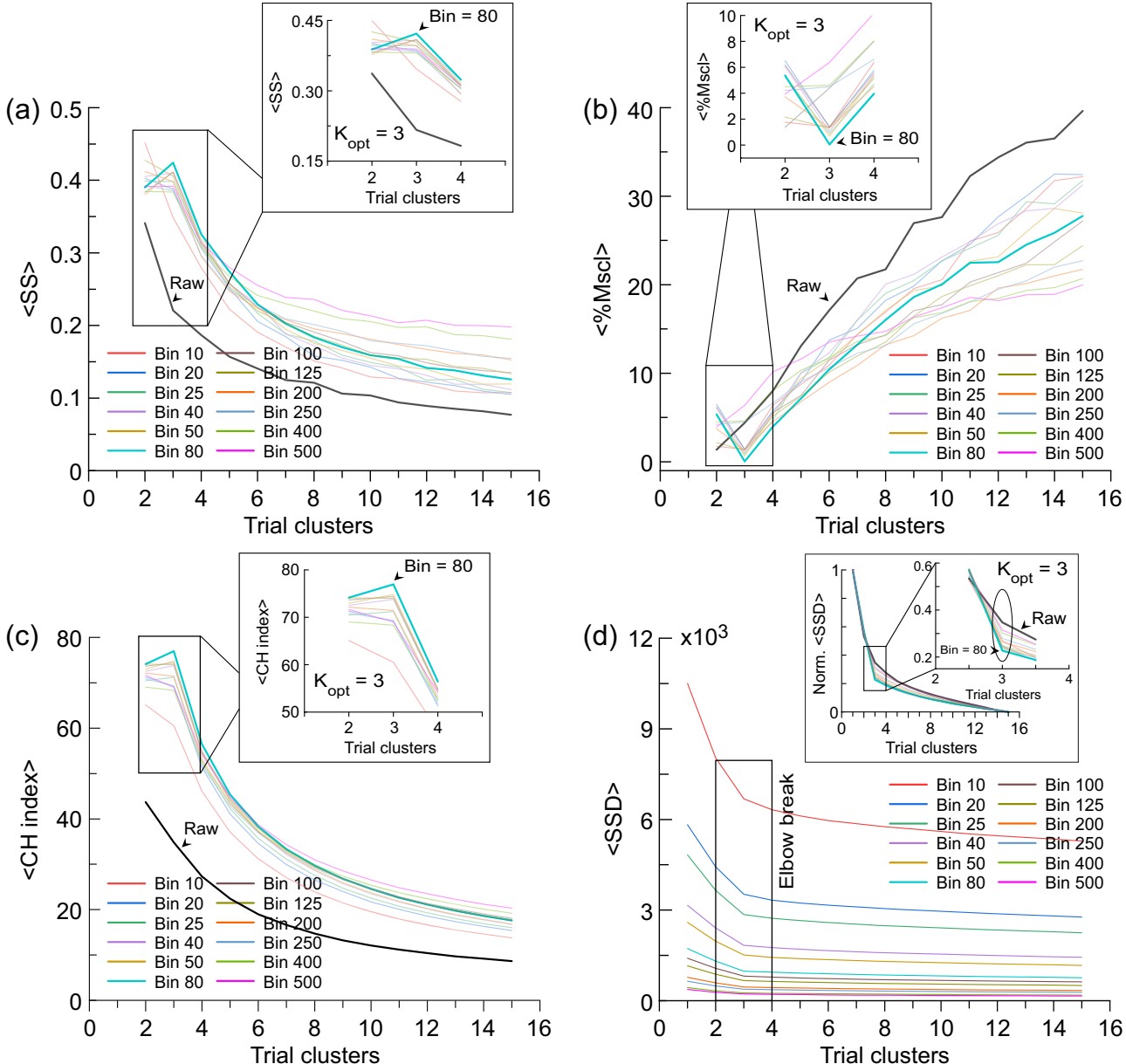

**Fig. 2 | Estimation of the optimum number of clusters ($K_{opt}$). a–d** The <SS> , <% Mscl> , <CH index> , and <SSD> metric profiles, considering 100 iterations of the code, are calculated for both unprocessed and the binned (by frames) PL traces following data normalization. Metric profiles are calculated for the data binned by 10, 20, 25, 40, 50, 80, 100, 125, 200, 250, 400, and 500 frames and compared with that for the pristine (raw) blinking trajectories (in black). The ideal window of binning is estimated at 80 frames, with the corresponding average metric profiles represented in cyan, as highlighted in each panel. The insets display zoomed view of the average metric profiles with the $K_{opt}$ indicated therein. Average metric profiles for the raw traces (in black) and optimally binned traces (binned by 80 frames, in cyan) are kept in bold.

multi-state blinking profiles of identical dimensionality (10,000 data-points) are deliberately introduced within the existing ensemble of PBB, TSB, and PEB trajectories. For this new dataset of 235 PL traces (unprocessed), the <SS> profile reveals an elbow at cluster 3 ($K_{opt}$) which is however not clear from the <CH index> , <SSD> , and <% Mscl> profiles (Supplementary Fig. 27). The <%EM> exhibits an abrupt rise at cluster 3 within the range SS > 0.2 to SS < 0.6 (Supplementary Fig. 28), implying the existence of three intersecting datasets. Considering $K_{opt}$ = 3, the clusters of raw trajectories demonstrate significantly misclassified blinking traces (Supplementary Fig. 29). Even upon preprocessing of PL trajectories with WL6 and binning by 80 frames, as optimized in the earlier sections, the existence of four sub-categories of blinking remains elusive. Rather, the V&L module suggests $K_{opt}$ = 3 in each case (Supplementary Figs. 30–31), and K-means

provides misclassified MSB trajectories throughout the clusters (Supplementary Figs. 32–33). In fact, manual consideration of $K_{opt}$ = 4 in the K-means algorithm has been incapable to segregate the MSB behavior (Supplementary Figs. 34–36), considering both the raw and processed datasets. This indicates that the MSB profiles are distinct, resembling neither themselves nor traditional blinking patterns, and can be considered as potent outliers.

Here, we attempt to distinguish MSB patterns from the mainstream PL traces, depending on β values of the power-law fitted PSD distributions. As MSB often entails multiple intensity levels with defined amplitudes, it is likely that few deep trap states contribute to these fluctuations. Consequently, a higher β value is anticipated according to MRC model[42–45]. We find β = 1.24 for the representative MSB trajectory (Supplementary Fig. 37a,b), and the <β> for this sub-

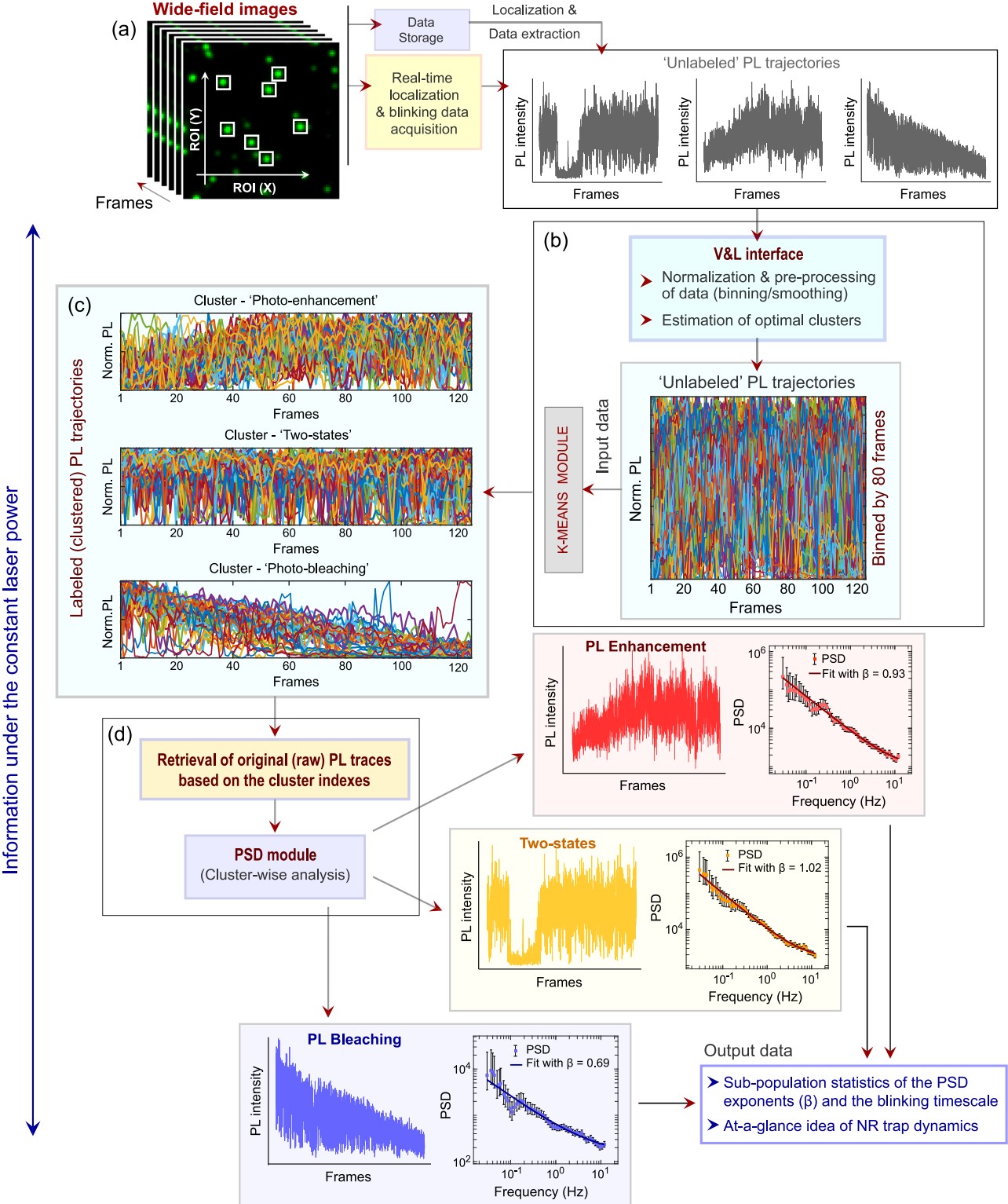

**Fig. 3 | Sequential schematic of the UML-PSD process. a** The wide-field PL images of the ROI, and automated (or manual) extraction and accumulation of blinking trajectories from spatially-localized diffraction limited emitters. **b** The subsequent V&L interface optimizes the level of data pre-processing (here binning) based on selected statistical metrics and determines the $K_{opt}$ for the ensemble. **c** Depiction of the UML (K-means) module where the processed (binned) data and the estimated $K_{opt}$ are fed into, and the individual clusters are constructed. **d** The subsequent PSD module retrieves the original blinking trajectories depending on the cluster indexes and performs cluster-wise PSD analysis on the raw data. The PSD distributions (with standard errors) are fitted with the power-law model to extract the exponent ($\beta$) values. The PSD distributions of three representative blinking traces from each cluster, along with the corresponding $\beta$ values, are shown in the subpanels.

category is usually higher as compared to the others. This provides an opportunity to differentiate the outlier MSB trajectories to some extent, which warrants further investigation. In this context, a supervised machine learning based classification of blinking signals, instead of clustering, could be a potential alternative. However, it requires a substantial number of training datasets comprising various blinking patterns, which is yet to construct using the existing domain knowledge.

### Assessment of synchronously blinking crystal grains

The unsupervised clustering algorithm further showcases application in identifying the crystal grains within the assembly (thin-film) that exhibit spatially synchronous PL blinking – a technique to measure the extent of communication between photoexcited distal charge carriers[16–18,55–57]. This communication relates to the mobility and diffusion length of the carriers, which can extend up to several microns for MHP systems[18,60]. Therefore, spatial synchronicity of PL blinking (or lack of) can unravel (dis)similarity between grains in terms of (in)homogeneous crystal orientation, energy landscape (traps) and the nature of grain boundaries. Typically, the spatial synchronicity of blinking is quantified by Pearson's correlation coefficient (PCC, Supplementary Note 8) between the PL trajectories of individual pixels as a function of distance. The analysis generates a correlation map of the assembly where each pixel demonstrates the corresponding PCC value[18,77,78]. Therefore, higher pixel values throughout distal nanodomains indicate a longer diffusion length of carriers, either in presence of fewer NR traps or permeable grain boundaries, suggesting excellent optoelectronic quality of the material (thin film). Recently, spatially correlated blinking has been utilized as a tool to investigate the carrier mobility within an assembly of Cl-treated MAPbI$_3$ crystals and thin-films (Supplementary Fig. 38a)[16]. The corresponding correlation maps (Supplementary Fig. 38b) revealed nanodomains with comparable (or contrasting) pixel-wise PCC values, inferring spatially diverse photophysics.

Nonetheless, PCC depends on two factors, (i) the trend (gradient) and (ii) the noise (fluctuation) of the signal. Therefore, two crystals undergoing different PL fluctuations, however, a similar intensity trend (PEB or PBB), can even show a significant correlation (Supplementary Fig. 39). Moreover, temporal similarity in PL trajectories can also lead to high intercorrelation as observed for the representative crystals 1 and 2 (PCC = 0.83) in Supplementary Fig. 38c,d. Hence, the pixel-wise PCCs become comparable, and grains become challenging to distinguish from the correlation map (Supplementary Fig. 38b). In such cases, two adjacent crystal grains can be misclassified as a single grain. This could bias the information on spatially (in)homogeneous charge carrier dynamics, indicating a limitation and lack of resolution in correlation (covariance)-based deterministic analyses.

Here, the K-means algorithm can be useful to spatially cluster the pixels depending on minimum Euclidean (point-to-point) distance between corresponding PL blinking trajectories. The UML is applied on the crystal assembly shown in the Supplementary Fig. 38a[16], where a time-averaged image (Fig. 4a) of the ROI is initially constructed. Next, the relevant pixels are selected above an intensity threshold (noise count, 200 cts) and background pixels are set to zero. The PL trajectories (6000 dimensionality) of selected pixels are extracted from the blinking movie (.tiff file) and the input dataset of PL trajectories is structured. This treatment reduces the participation of unwanted pixels and background noise during the clustering process. The V&L interface bins the input trajectories by W = 10, 20, 25, 30, 40, 48, 50, 60, 75, 80 and 100 frames (equivalent to W×100 ms time-bins), and determines the optimum binning window as well as the $K_{opt}$ for the ensemble. This estimation is performed relying on the <SS> and <% Mscl> metrics (Fig. 4b), which are proven beneficial in the earlier sections. The <SS> profiles show a declining trend and an elbow break

at cluster 3 ($K_{opt}$) for both the original and binned PL trajectories. The declining trend owes to the similar blinking propensity of crystal 1 and 2, as already evident from the correlation maps (Fig. S28c). The <% Mscl> profiles suggest the optimum binning window at 40 frames that exclusively show minimum misclassification at cluster 3. Afterward, blinking data binned by 40 frames are subjected to the K-means module and the pixel-wise blinking characteristics are clustered. The cluster-map (Fig. 4c) of pixels represents three crystal grains at their respective spatial locations. It is worth noting that clustering of the unprocessed blinking traces, considering $K_{opt}$ = 3, incorporates misclassified pixels as observed in Supplementary Fig. 40. This further highlights the relevance of data smoothing and dimensionality reduction to achieve a precise cluster-map.

Next, the original blinking trajectories are retrieved based on the cluster indexes of pixels (Fig. 4d) and the PSD distributions of pixelwise original PL trajectories are constructed. The power-law fitting parameters, for each pixel, are utilized to generate β-map of the entire assembly. Subsequently, the <β> value for each clustered domain is calculated. The cluster-map and β-map of the crystal assembly collectively offer valuable insights into the charge carrier communication, grain boundaries and spatially varied nature of the NR traps. A decrease in pixel-wise β values for the C3 domain suggests relatively lower contribution of high amplitude PL fluctuations (Fig. 4d), implying a lack of active deep NR traps in C3, in contrast to the C1 and C2 domains. The entire UML-PSD process, including data extraction, optimization of binning window, determination of the $K_{opt}$ and K-means clustering, PSD analysis and reconstruction of cluster-map and β-map, typically requires a few minutes timescale, considering wide-field PL movie file of dimension ~3 × 3 μm$^2$, comprising 400 pixels and 6000 dimensionality of blinking data per pixel.

## Discussion

We present a straightforward 'clustering-segregation-analysis' (UML-PSD) methodology for fast-track optical characterization of the semiconductor NCs through wide-field microscopy. Essentially, the algorithm clusters the major PL blinking patterns in an ensemble, delivers subpopulation statistics and provides class-wise PSD information that unveils insights on diverse charge carrier dynamics. The clustering technique further shows utility in distinguishing the synchronously blinking grains within a crystal assembly which infers on the carrier diffusion through grain-boundaries. Simultaneously, the β-map of grains reveals immediate insights into spatially varied nature of traps. We find that the metrices like SS and %Mscl are perhaps more useful to estimate optimum number of clusters ($K_{opt}$) in microscopy generated PL blinking data as compared to the CH index and SSD techniques. Furthermore, data preprocessing ensures accurate determination of the $K_{opt}$, improves clustering accuracy, and reduces the analysis time. The UML module additionally proves versatility across multiple domains of (micro)spectroscopy, for instance, we demonstrate an easy way to segregate (un)biased STS signals (at room temperature) from SAMNs (see Supplementary Note 9), which is often challenging at solid-liquid interfaces. We have further tested the potential of the UML methodology to investigate the effect of ligand-based (didodecyldimethylammonium bromide, DDAB) surface passivation on the stability and blinking characteristic of pristine CsPbBr$_3$ nanocrystals, in ambient conditions (see Supplementary Note 10). However, our laptop-based UML-PSD analysis sometimes underperforms due to computational hurdles in managing large datasets with high dimensionality. We experienced such limitations during an intensity-wise clustering of nano- or microdomains exhibiting apparently similar PL fluctuations within a bulk CsPbBr$_3$ crystal, which spans approximately 18 × 18 μm$^2$ and comprises roughly 23,500 pixels with PL blinking trajectories of 6000 dimensionality (see Supplementary Note 11). A graphics processing unit (GPU) could be useful to enhance the performance of the UML-PSD module.

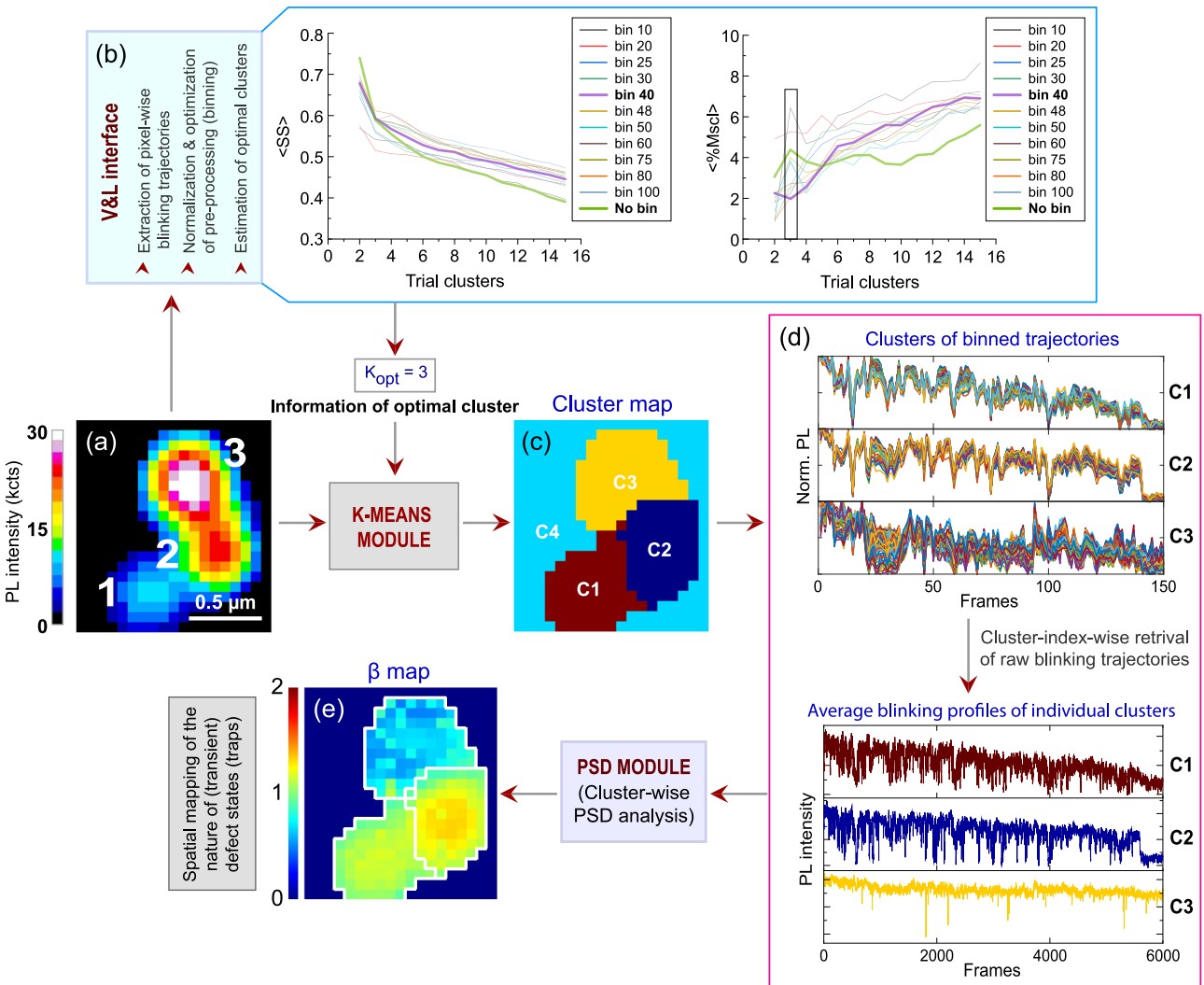

**Fig. 4 | The UML-PSD workflow for crystal assemblies. a** The time-averaged image of a PL movie (.tiff), recorded for three self-assembled MAPbI$_3$ crystals, with background pixels set to zero. **b** Illustration of the V&L interface that extracts and normalizes the unlabeled blinking traces corresponding to relevant pixels. Calculation of the <SS> and <%Mscl> profiles for the normalized ensemble of PL trajectories determines the $K_{opt}$ at cluster 3 and estimates the optimum binning window at 40 frames. Metric profiles for the raw traces (in green) and optimally binned trajectories (in violet) are shown in bold. **c** Schematic of the segregation (UML-PSD) module, where the UML section applies the K-means algorithm to cluster the pixels exhibiting analogous blinking behavior, generating a cluster map of the crystal assembly based on the ensemble of binned PL trajectories. **d** The initial stage of the PSD module retrieves the raw PL trajectories depending on the cluster indexes. **e** The subsequent part computes the PSDs for the labeled blinking traces, extracting the pixel-wise power-law exponents (β) to construct the corresponding β-maps for the clusters.

In future, the methodology can be upgraded to a data-driven supervised deep-learning (neural network) module combined with PSD analysis to classify and interpret the PL blinking signals. The data driven module could enhance accuracy in identifying the outlier (e.g., MSB) blinking features. Moreover, super-resolution techniques could be integrated upstream of the machine learning module to enable extraction and classification of time series with precise spatial localization. Such an approach would be particularly valuable for complex biological images, offering a pathway to extend our methodology beyond photo-physics of nanocrystals toward broader interdisciplinary applications. Like existing threshold-dependent data analysis software[79], we aim to integrate these automated data analysis methodologies into the data acquisition interface of conventional optical microscopes. Particularly, automated data analytics combined with the upcoming technology like lens-free fluorescence microscopy[80], can reduce both the instrument volume as well as the data analysis time. Realization of such futuristic smart microscopes will enable hassle-free and fast optical assessment of nanocrystals (batch-wise) – a pathbreaking advancement highly desired in nano-semiconductor-based industries as well as academic research.

## Methods

### Synthesis of cesium oleate precursor

0.4 g of Cs$_2$CO$_3$ was loaded with 15 mL of 1-octadecene (ODE) and 1.725 mL of oleic acid (OA) in a 100 ml three-neck round-bottom flask. The reaction mixture underwent vacuum degassing for 1 h at 120 °C to remove volatile impurity and to facilitate the formation of Cs-oleate complex. The flask was filled with N$_2$. Subsequently, the solution mixture was further annealed for 1 h at 120 °C to ensure the formation of a clear Cs-oleate solution. The resulting solution was collected under elevated temperature conditions and stored in a vial filled with nitrogen. Upon cooling to room temperature, the mixture solidified, requiring the cesium oleate to be heated above 80 °C for liquefaction before further use.

## Synthesis of CsPbBr$_3$ NCs

In a standard synthesis procedure[30], a 50 mL three-neck round-bottom flask was charged with 89.2 mg of lead oxide, 238.8 mg of phenacyl bromide, 2 mL OA, and 10 mL ODE. The mixture underwent evacuation and nitrogen purging under stirring at room temperature for three times. Subsequently, the entire system was degassed with nitrogen at 120 °C for 1 h. Once all the salts were dissolved, the reaction temperature increased, and at 220 °C, 1 mL oleylamine (OAm) was injected. The solution initially turned red within a minute, gradually shifting to yellow after about 20 min. At 220 °C, 0.8 mL of pre-synthesized Cs-oleate was rapidly introduced into the clean yellow solution and annealed for 5 min, yielding dodecahedron CsPbBr$_3$ nanocrystals. Samples were collected promptly using ice quenching. To purify the final product, crude reaction mixtures were mixed with 30 mL of methyl acetate in a 50 mL centrifuge tube followed by centrifugation at ~2285 × $g$ (8700 rpm, revolutions per minute) for 10 min. The supernatant was discarded, and the remaining precipitate was redispersed in 4 mL hexane for subsequent use.

Similar shape and dimension of the crystals (21.4 nm ± 2.57 nm) can be confirmed from transmission electron microscopy (TEM) images and size distribution of the CsPbBr$_3$ NCs (Supplementary Fig. 9a,b). The phase purity is established by the X-ray diffraction (XRD) patterns which are identical with the standard XRD pattern (Supplementary Fig. 9c) of the orthorhombic phase of CsPbBr$_3$ crystals.

## Preparation of bulk CsPbBr$_3$ microcrystal

74.5 mg CsBr and 128.5 mg PbBr$_2$ were dissolved in 1 mL anhydrous DMSO by stirring at room temperature to prepare 0.35 M CsPbBr$_3$ precursor solution. Subsequently, the as-prepared 10 μl solution was placed in between two freshly cleaned glass coverslips in a sandwich manner, followed by annealing on a hot plate at 70 °C for 10 minutes and the temperature was increased to 150 °C for another 10 min. Upon cooling, cover slips were separated carefully for further use in wide-field optical imaging. The time-averaged (300 s) wide-field image of the bulk crystals is demonstrated in Supplementary Notes, section 11.

## Preparation of MAPbI$_3$ crystals

The synthesis protocol of the (Cl-treated) MAPbI$_3$ MCs can be found in the literature *Nanoscale* **15**, 5437–5447(2023)[16]. In brief, the precursor solution is prepared by dissolving lead iodide (PbI$_2$, 0.04 mmol, 18.8 mg) and methylammonium iodide (MAI, 0.12 mmol, 18.8 mg) in 20 mL of acetonitrile (ACN, 99.8%). Simultaneously, the capping agents, 72 μL of OAm and 60 mg of trioctylphosphine oxide (TOPO), are dissolved in 30 mL toluene as antisolvent and divided into six batches after stirring. Five minutes after injection of 1.3 mL of the precursor solution into the vigorously stirred capping agent solution, an extra 7 mL of toluene is added dropwise. The mixture is kept stirring at room temperature for 4 h at 600 rpm, resulting in a dark brown suspension containing MAPbI$_3$ nanocrystals. The dark brownish suspension is then washed twice with toluene after centrifugation for 30 minutes at ~533 × $g$ (4200 rpm). In order to obtain a substantial yield for optical measurements, six batches are mixed, divided into 4 batches and re-dispersed in 4 mL of toluene. Methylammonium chloride (MACl, 13.6 mg, 200 μmol) is dissolved in 0.5 mL absolute ethanol. The 0, 3, 9, and 17 molar% (Cl: I molar ratio) Cl post-treatment is realized by directly adding 0, 1, 2.5 and 5 μL of the MACl solution (in ethanol), respectively, to 4 mL of the MAPbI$_3$ crystal suspension (in toluene) resulting in ethanol/toluene (v/v) ratios of 0%, 0.025%, 0.065% and 0.125%. The mixture is then kept stirring for 30 min at 600 rpm. The suspension is then washed again by centrifugation for 30 minutes at ~533 × $g$ (4200 rpm), and the residue is re-dispersed in 4 mL of toluene. The final crystal suspensions are kept in the dark. For wide-field microscopy, 20 μL of the nano/microcrystal suspension in toluene is dropped and spin-coated at 1000 rpm for 60 s on a clean glass coverslip.

## Preparation of SAMNs

The SAMNs of the N,N′-bis(n-alkyl)-naphthalenediimides (NDI) are prepared following the method mentioned in *Chem. Mater.* **33(22)**, 8800–8811(2021)[58]. Briefly, n-octadecylamine (compound 1) is reacted with 1,4,5,8-naphthalenetetracarboxylic dianhydride (compound 2). The condensation is sped up using a microwave reactor. Compound 2 is used in excess to form predominantly octadecyl naphthalene monoimide (NMI) and reduce the formation of double alkyl functionalized R-NDI. During the work-up, unreacted 2 was solubilized in water by hydrolysis in an aqueous NaOH solution (1 M). Insoluble NMI was isolated by filtration and dispersed in an aqueous hydrochloric acid solution (3 M) to protonate the carboxylic acid groups. Protonation was required to increase the electrophilicity of the compound for the subsequent condensation reaction. Dried NMI was reacted with aliphatic α,ω-diamines (ranging from 1,3-diaminopropane to 1,12-diaminododecane) using a microwave reactor. All double NDIs were purified by recrystallization from chloroform or toluene and obtained as crystalline compounds in medium to high yields (34-81%).

## Thin-film preparation of CsPbBr$_3$ NCs and Wide-field PL imaging.

The dilute solution CsPbBr$_3$ NCs (in hexane) was sonicated for 1-2 minutes at 37 KHz, and spin-coated on the glass coverslips (24×24 mm$^2$) at 2000 rpm to deposit the thin-film. The epifluorescence imaging of CsPbBr$_3$ NC thin-film was conducted using wide-field mode and an objective lens with 1.45 numerical aperture, 60X magnification. The NCs were excited by a 488 nm continuous wave laser with a power density of 44.2 mW/cm$^2$ (after objective lens), while the emission was detected in wide-field mode, through a combined 505 long-pass and 488 nm-561 nm bandpass filters. The 16-bit images were collected using a Hamamatsu EM-CCD digital camera (model number C9100-13, quantum efficiency >90%), at a shutter speed of 33.3 Hz (30 ms temporal resolution). The blinking data acquisition time was set at 300 s to accumulate 10,000 PL intensity data points. The entire microscopy was performed at room temperature.

## Data extraction.

Initially, background flattening was performed for each frame of the PL movie, using ImageJ software, and a time averaged frame was obtained. On this time averaged image, the regions (pixels) corresponding to the clusters/aggregates were manually identified, cropped and eliminated. Next, the ImageJ software allowed to find the location of intensity maxima for the rest of the spatially-segregated diffraction limited spots. This produces a list of (x,y) location coordinates. These coordinates are considered as the center pixel of a 3×3 pixels area (diffraction limit) where nine pixels are set as [(x-1,y + 1), (x, y + 1), (x + 1, y + 1); (x-1,y), (x, y), (x + 1, y); (x-1,y-1) (x, y-1) (x + 1, y-1)]. Thereby, the intensity (blinking) profile considering these nine pixels was extracted using MATLAB. In the entire process, manual intervention was considered only for the step associated with the selection and elimination of unwanted pixels to achieve relevant scientific information.

## Scanning tunneling spectroscopy of the SAMNs.

The samples for the scanning tunneling microscopy (STM) measurements were prepared under ambient conditions by applying a drop of the saturated solution of a given double NDI derivative onto a freshly cleaved highly-ordered-pyrolytic-graphite (HOPG) surface (grade ZYB, Advanced Ceramics Inc., Cleveland, Ohio, United States) followed by creating a solution flow by touching one corner of the HOPG sample with a clean Kimwipe tissue paper after which the sample was heated at 100 °C for 10 min. This annealing step led to the evaporation of the solvent. Hence a drop of neat solvent was applied to the surface prior to imaging. The STM experiments were carried out using a PicoSPM (Agilent) operating in the constant-current mode with the tip immersed in the solution at room temperature. The STS data (I-V characteristics) was collected upon scanning an area of 20 × 20 nm$^2$ with sub-molecular

resolution. As the fast-scanning axis reached the 'y' values of 5, 10, and 15 nm, imaging of the area was stopped while the tip was kept in feedback. Upon recording the I-V spectra, the feedback was briefly interrupted while 5 sweeps were recorded. This was followed by a continuation of imaging the area to ensure the tip resolution, and thus the tip shape did not alter due to the STS data acquisition. Upon measurement, the tunneling junction was kept at a tunneling current of 0.5 nA and within a voltage bias of −1.0 V to −1.5 V. All data were collected at room temperature (295 K) under the ambient atmosphere.

**Visual & Logical' (V&L) module.** The Visual & Logical module consists of MATLAB codes to calculate three commonly explored statistical metrics: (i) Silhouette score (SS)[61–64], (ii) Calinski-Harabasz (CH) index[65–67] and (iii) Sum square deviation (SSD, Elbow method)[68–70] (Supplementary Notes 4–6). These metrics are calculated for a range of trial clusters ($N = 2$ to 15) and plotted against the increasing number of N values. Subsequently, it displays the metric profiles, and the optimum number of clusters ($K_{opt}$) can be determined by inspecting these profiles which either exhibit an elbow (e.g., first drastic change of slope at a certain point) or a maximum value at $K_{opt}$ for the ensemble. For instance, the SS profile typically decreases with $N$ and show an elbow at $K_{opt}$ when signals consist similar patterns yet different ranges of values. Contrastingly, the SS profile undergoes a maximum value at $K_{opt}$ when contrasting patterns are present within the ensemble. The conditions are briefly discussed in the Supplementary Note 4. The trial cluster corresponding to the elbow break, or the maximum value of metric profiles can be visually/logically identified, which provides an opportunity to automate the decision on $K_{opt}$ for the K-means algorithm. However, these scenarios can alter depending on the dimensionality of data and the extent of preprocessing (smoothing/binning) of the signals, as seen for the high-dimensional PL blinking signals. Eventually, the $K_{opt}$ ensures maximally aligned signals within the individual clusters and marginal resemblance between neighboring clusters.

Although the mathematical limit of the SS value is −1 to +1, and 0 to ∞ for both the CH index and SSD, the metrices show slight variation within the range upon each iteration of the code. This is likely due to randomly chosen initial centroids by the algorithm. To mitigate this discrepancy, we execute each code on the same dataset for 100 iterations (epochs) and consider the mean profiles of the Silhouette score (<SS>), CH index (<CH index>), and SSD values (<SSD>) for the trial clusters. Afterward, the $K_{opt}$ is determined according to characteristic features of these average metric profiles. The 'Visual & Logical' (V&L) interface graphically displays the metric profiles and provides an opportunity to visually/logically estimate the $K_{opt}$ value. A visual inspection of the metric profiles allows manual determination of $K_{opt}$, which may sometimes be advantageous to align with the scientific explanations according to the domain knowledge. All analyses are performed using an 11th Generation laptop with Intel(R) Core (TM) i7 processor and 16 GB RAM configurations.

## Data availability
The trial blinking datasets, with and without multi-state blinking trajectories, can be found in the repository https://doi.org/10.5281/zenodo.17715957[81].

## Code availability
The codes are stored in the repository https://doi.org/10.5281/zenodo.17715957 (for UML analysis[81]) and https://zenodo.org/records/8027381 (for PSD analysis[82]). These codes can be universally utilized on any dataset of choice according to the application and interest of the user.

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

## Acknowledgements

R.R., K.S.M. and S.D.F. acknowledge financial support from the Research Foundation – Flanders (FWO grant numbers G0H2122N, EOS 40007495) and the KU Leuven Internal Funds (grant number C14/23/090). B.P. acknowledges Industrieel Onderzoeksfonds KU Leuven (IOF)-VTI-25-00160. S.S. acknowledges the support of Marie Skłodowska-Curie postdoctoral fellowship (No. 101151427, SPS_Nano) from the European Union's Horizon Europe program, short stay abroad grant (K257023N) and travel grant (K147824N) from Research Foundation-Flanders (FWO).

J.H. acknowledges financial support from the Research Foundation-Flanders (FWO grant numbers S002019N, G098319N, S004322N Gigapixel and G0AHQ25N), the KU Leuven Research Fund (iBOF-21-085 PERSIST), the Flemish government through long-term structural funding Methusalem (CASAS2, Meth/15/04), and the MPI as a fellow. E.D. acknowledges funding from the KU Leuven Internal Funds (grant numbers C14/23/090, CELSA/23/018) and FWO grant number G0AHQ25N, and the European Union (ERC Starting Grant, 101117274 X-PECT). However, the views and opinions expressed are those of the authors only and do not necessarily reflect those of the European Union or European Research Council. Neither the European Union nor the granting authority can be held responsible for them.

## Author contributions

Conceptualization: A.M., E.D., Methodology (Machine Learning): A.M., T.B., Methodology (Power spectral density): S.S., Wide-field imaging: A.M., L.C., Scanning tunneling spectroscopy: R.R., Sample preparation & characterization (CsPbBr$_3$ system): B.P., H.J., C.S., M.P., L.L., Sample preparation & characterization (Alkyl-NDI SAMNs): R.R., A.T.R., Investigation & Visualization: A.M., Funding acquisition: E.D., J.H., Supervision: E.D., J.H., Writing – original draft: A.M., R.R., Writing—review & editing: K.S.M., S.S., B.P., S.D.F., M.R., E.W.M., J.H., E.D., #R.R. and B.P. contributed equally.

## Competing interests

The authors declare no competing interests.
