## [Transparent Peer Review file · Nature Communications]

Machine Learning for Microscopy Data Analytics Targeting Real-time Optical Characterization of Semiconductor Nanocrystals

Corresponding Author: Professor Elke Debroye

Version 1:

Reviewer comments:

Reviewer #1

(Remarks to the Author)

The authors develop a machine-learning-assisted segregation and analysis method of large PL trajectories datasets, that circumvents the non-trivial issue of subjective threshold-dependent analysis. They use it to investigate carrier dynamics and nature of trap states in large sets of heterogeneous blinking traces, demonstrating its predicting power and discussing its applicability to various landscapes (such as the application to the assessment of synchronous blinking in crystal grains or the analysis of STS-generated I-V curves), and its limitations.

The authors do a great job explaining in great detail all the phases of the process using excellent schematics to convey complex concepts in an intuitive fashion.

This is a very promising automated analysis method of large datasets that can be performed in nearly real-time and has wide applicability. It will undoubtedly be of interest to the wide readership of Nature Communication. I therefore recommend publication after the following minor comments have been addressed:

1) On page 2 the authors state that “Therefore, understanding the nature and density of NR traps is crucial to qualitatively evaluate these nanomaterials...”. The method presented, however, does not seem to be able to estimate the density of traps, as the value of the exponent Beta only provides information on the depth (i.e., the nature) of the traps, not on their density.

2) The presence of unnumbered Figures in the SI within the Supplementary notes can be misleading when the authors, in the main text, refer to numbered SI Figures. I suggest naming the former as “Diagram” or “Schematic” or something else - as long as it is not “Figure” - (and number them), and keep the latter as numbered Figures. In this way the reader won't mistake “Supplementary Fig. n” with the n-th figure shown in the supplementary material.

3) The acronym “ROI” is never defined.

4) On page 5, the authors mention a metric which “measures the percentage of misclassification (%MscI) by segregating those blinking traces which exhibit a negative SS value ($SS < 0$)”. However, there are no negative SS values in SI Fig.3a ...

5) On page 5, the authors claim that they “find a sharp rise of $\langle \%EM \rangle$ at cluster 3, which is absent below 0.2 ($SS < 0.2$). Therefore, the span of intersection can be anticipated to be 0.2 in Silhouette space, while elements outside the intersection ($SS > 0.2$) are well-segregated among three blinking categories.” However, it seems to me that the sharp rise they mention is also visible at $SS < 0.1$ for the same cluster 3. So, it is difficult to objectively determine a value for the span of intersection in Silhouette space.

6) In the SI, section 5, page 6: shouldn't B(k) be the inter- (not “intra”) cluster dispersion?

7) The definition of “epoch”(= code iteration?) used in many figure labels in the SI is missing.

(Remarks on code availability)

Reviewer #2

(Remarks to the Author)

Debroye et al. present a manuscript describing an unsupervised machine learning algorithm to analyze the intensity fluctuations of nanocrystals recorded through widefield fluorescence microscopy. The algorithm and the mathematical tools utilized in the analysis are robust and there is no doubt that this methodology excels in the field and is of great intellectual value. It is, however, a manuscript focused solely on the algorithm, and I believe that for Nature Communications a broader application should be presented to be considered for publication. The main application of the algorithm in the PL datasets is that it accurately categorizes them into three categories: a two-state, a PL enhancement and a PL Bleaching. Two-state blinking, PL enhancement (photobrightening) and PL bleaching (photodarkening) are processes well studied in the literature and are not new (J. Chem. Phys. 115, 1028–1040 (2001), Topics in Current Chemistry 2016, 374 (5), 58. Section 3.4). Furthermore, simpler algorithms have been established, not only to determine the behavior of the on-off trajectories, but for fluorescence lifetimes and spectral diffusion (J. Phys. Chem. A 2020, 124, 3494–3500, Phys Rev B 2014, 90 (19), 195302.) The authors should make an effort to provide -new- knowledge in the field of photophysics (or STM) while using the algorithm, only then this manuscript is worthwhile of Nature Communications. I believe that addressing the comments below could strengthen the impact of the manuscript.

1. A blinking process is the fingerprint of a single nanocrystal or molecule. How do the authors make sure that the multi-state-blinking (on and off) is the result of a single nanocrystal and not the of two or more NCs in the same diffraction limited spot? Although a second order correlation measurement (antibunching) is the definitive proof, the authors must provide clear evidence that the trajectories are indeed a single NC. Or, on this same train of thought, is the algorithm capable of categorizing the blinking trajectories and assigning them to two or more NCs in the same diffraction limited spot?
2. While selecting the pixels from the .tiff file for the blinking trajectories, the authors claim that they were carefully selected after background subtraction with the use of ImageJ. Although it is not mentioned, I believe that this procedure is human-driven and may bias the subsequent analysis. I am wondering if the authors thought in applying a spatial analysis of the photons, similar to super resolution microscopy, to accurately and automatically assign the pixels that could correspond to a single NC. If the authors could bridge this new analysis to the one widely employed in super resolution microscopy it would be of great value, not only for the photophysics and physical chemistry fields, but also for the biological sciences.
3. How does the analysis look like without normalizing the intensity trajectories, is there something that the algorithm can tell us about the heterogeneity in terms of brightness of the individual NCs?
4. The time resolution (30ms) of the frame is defined by the end of the manuscript. The authors should use time and time-binning vocabulary throughout the manuscript and not frame-binning to make a closer connection to the photophysical processes well known in the literature.
5. It is well known that different excitation fluences affect the photophysical behavior of the NCs (Nature 1996, 383 (6603), 802–804). The authors should provide analysis of the same sample, measured at different fluences, and demonstrate that the algorithm is capable of distinguishing and classifying the trajectories with significant differences.
6. Besides the “normal” blinking, the PL enhancement and PL bleaching, is the algorithm capable of determining photobleaching? i.e. QDs that were showing the on-off states and at certain time point it did not recover from the off state?
7. Authors should provide more details of their detection scheme for PL measurements, that is, camera quantum efficiency, brand and model.

Minor comments.

Fig 1. The text font inside the figure is too small.

(Remarks on code availability)

There was no code in ESI.

Reviewer #3

(Remarks to the Author)

This work proposes the use of machine learning algorithms for the optical characterization of the “quality” of all-inorganic perovskite NCs via their emitting blinking behavior. At least for II-VI semiconductor NCs the blinking behavior together with the profile of the excited state (PL lifetime) have been used to assess the “quality of these NCs, a good NC exhibiting a two-state on-off blinking with a single exponential lifetime. Perovskites NCs are a little more challenging in the blinking behavior with existing reports showing blinking even from micron sized crystals (see the works of Scheblikyn in NanoLetters 2015) , and this makes the proposed assessment challenging, at least when it comes to judging the “good” optical quality of a perovskite compound, either NC or microcrystal. Because of this, I commend the authors for the grand challenge they have taken. I also think that the use of power spectral density (PSD) in calculating the blinking behavior is preferred against other methods, in particular when dealing with such heterogeneous NCs, see the work of Pelton (APL 2004) who demonstrated the use of PSD to uncover blinking even in ensembles of emitters in solution and not necessarily isolated NCs. So the proposed method of Hofkens and team is interesting as it extends to ensemble of emitters, which is done here to a certain extent by correlating synchronous blinking from multiple grains in a crystal. Having the ability to classify charge carrier dynamics with spatio-temporal accuracy is important to the community and to be able to classify traps in perovskites is important as well, and for large samples this involves large data sets where ML can help. Also the synchronous blinking among traps in a single crystal is a nice demonstration of the capability of this ML method. All together I find the study interesting and innovative. The authors could comment how their method could be expanded to solution systems where blinking is also present although spatial info may be lost.

(Remarks on code availability)

Version 2:

Reviewer comments:

Reviewer #1

(Remarks to the Author)

The authors have satisfactorily addressed all of my comments. I therefore recommend publication of the manuscript in Nature Communications.

Marco Califano

(Remarks on code availability)

Reviewer #2

(Remarks to the Author)

Comments.

Debroye et al. present a revised manuscript that effectively addresses the reviewers' concerns and demonstrates significant improvement. I commend the inclusion of analyses on unnormalized data sets and nanocrystals with varied passivation ligands. However, I find a contradiction in the authors' rebuttal letter that warrants clarification.

The authors' response addressing the general comment 2 states: "Our contribution lies in providing a universal and threshold-independent solution. The unsupervised machine learning-power spectral density (UML-PSD) methodology efficiently segregates diverse blinking behaviours and delivers category-specific information on intensity fluctuations and the nature of trap states," Yet, in addressing comment 4, they acknowledge: "The algorithm can perform intensity-based clustering on the un-normalized blinking traces. However, it will lack efficiency in producing pure clusters, due to misclassifications...".

This discrepancy should be addressed. The authors ought to acknowledge in the manuscript that the algorithm's performance is limited when applied to less processed data. Specifically, it is unclear how the method can be considered threshold-independent if it struggles with unnormalized traces. Clarifying this point would strengthen the manuscript's claims and improve transparency.

Once this minor revision is incorporated, I believe the paper is suitable for publication in Nature Communications, and no further revisions are necessary.

(Remarks on code availability)

Coding and computer science is beyond my expertise.

Reviewer #3

(Remarks to the Author)

I am pleased with the answers provided by the authors.

(Remarks on code availability)

i am not familiar with such applications

Detailed point-by-point author response to referee's comments

We are grateful to the editor and the reviewers for the thorough evaluation and constructive feedback on our manuscript, “*Machine Learning for Microscopy Data Analytics: Toward Real-time Optical Characterization of Semiconductor Nanocrystals*” We appreciate that all reviewers recognized the novelty and broader significance of the study, as well as the rigor of our methods, data, and analysis, overall, that the work aligns with the journal’s scope (under the call “*Self-driving labs and automation software for chemistry and materials science*”) and high standards. At the same time, we respectfully acknowledge the concerns raised. To address all points comprehensively, we provide a detailed, point-by-point responses below.

Reviewer #1 (Remarks to the Author)

General Comments: The authors develop a machine-learning-assisted segregation and analysis method of large PL trajectories datasets, that circumvents the non-trivial issue of subjective threshold-dependent analysis. They use it to investigate carrier dynamics and nature of trap states in large sets of heterogeneous blinking traces, demonstrating its predicting power and discussing its applicability to various landscapes (such as the application to the assessment of synchronous blinking in crystal grains or the analysis of STS-generated I-V curves), and its limitations.

The authors do a great job explaining in great detail all the phases of the process using excellent schematics to convey complex concepts in an intuitive fashion.

This is a very promising automated analysis method of large datasets that can be performed in nearly real-time and has wide applicability. It will undoubtedly be of interest to the wide readership of Nature Communication. I therefore recommend publication after the following minor comments have been addressed:

Response: We sincerely thank the reviewer for positive evaluation of our manuscript, correctly understanding the workflow and highlighting the potential of our work that can be of interest to the audience of Nature Communication.

Comment 1: On page 2 the authors state that “Therefore, understanding the nature and density of NR traps is crucial to qualitatively evaluate these nanomaterials...”. The method presented, however, does not seem to be able to estimate the density of traps, as the value of the exponent Beta only provides information on the depth (i.e., the nature) of the traps, not on their density.

Response: We agree with the reviewer. The photo-bleaching or photo-brightening trend of a PL trajectory implies a temporal change in the density of nonradiative traps which is related to the degradation/improvement of the nanocrystal. However, it cannot unravel the overall density of traps at a certain instance.

Our unsupervised machine learning (UML) module performs segregation of these trends. Subsequently, power spectral density (PSD) analysis provides class-wise information on the β values. Altogether, the information of temporal stability (formation/annihilation of traps) as well as the nature (depth) of dynamic traps can be obtained for the same nanocrystal.

In agreement with the reviewer, we have now rectified the sentence on page 2 as, “**Therefore, understanding the nature of NR traps and temporal variation of their density is crucial to qualitatively evaluate these nanomaterials, especially for optoelectronic applications, which can be achieved by analysing PL blinking characteristics through optical microscopy techniques.**” (Highlighted in yellow; page 2 of the Revised Manuscript)

Comment 2: The presence of unnumbered Figures in the SI within the Supplementary notes can be misleading when the authors, in the main text, refer to numbered SI Figures. I suggest naming the former as “Diagram” or “Schematic” or something else - as long as it is not “Figure” - (and number them), and keep the latter as numbered Figures. In this way the reader won’t mistake “Supplementary Fig. n” with the n-th figure shown in the supplementary material.

Response: We thank the reviewer for this valuable suggestion. We have now addressed the unlabelled figures under the section “Supplementary Notes” as “Supplementary Diagram 1–8”. (Highlighted in yellow; page 2, 3, 4, 5, 9, 10, 11, 12 of the Revised Supplementary Information)

Comment 3: The acronym “ROI” is never defined.

Response: We have now defined “ROI” as the “region of interest”. (Highlighted in yellow; page 3 of the Revised Manuscript and page 8 of the Revised Supplementary Information)

Comment 4: On page 5, the authors mention a metric which “measures the percentage of misclassification (%MscI) by segregating those blinking traces which exhibit a negative SS value ($SS < 0$)”. However, there are no negative SS values in SI Fig.3a ...

Response: We appreciate the reviewer’s concern. We wish to clarify here that the Supplementary Figure 3a displays the SS profiles calculated for the entire ensemble of blinking trajectories. Each of the SS profiles is generated by a single iteration of the code (a total of 100 iterations) and thereby the mean SS profile (Supplementary Figure 3d) is achieved. By the definition of silhouette score, the overall SS for an ensemble is the average of the silhouette scores of all individual blinking trajectories. Thus, there can be misclassified blinking trajectories with $SS < 0$; however, the overall SS for the ensemble will remain a positive number. For this reason, negative SS values are not reflected in the Supplementary Figure 3a.

To address the reviewer’s comment, we slightly modified the sentence as, “To quantify this resemblance, we incorporate another metric into the V&L module which measures the percentage of misclassification (%MscI) by segregating the blinking traces that exhibit a negative SS value ($SS < 0$) within the ensemble.” (Highlighted in yellow; page 5 of the Revised Manuscript)

Comment 5: On page 5, the authors claim that they “find a sharp rise of $\langle EM \rangle$ at cluster 3, which is absent below 0.2 ($SS < 0.2$). Therefore, the span of intersection can be anticipated to be 0.2 in Silhouette space, while elements outside the intersection ($SS > 0.2$) are well-segregated among three blinking categories.” However, it seems to me that the sharp rise they mention is also visible at $SS < 0.1$ for the same cluster 3. So, it is difficult to objectively determine a value for the span of intersection in Silhouette space.

Response: We agree with the reviewer that the “sharp rise” depends on the analyst’s decision. Instead, we want to correct ourselves by stating “change in the slope”, which is perhaps a more appropriate term to inspect the $\langle EM \rangle$ profiles. We find the change of slope is prominent for the $\langle EM \rangle$ profile, considering $SS < 0.3$, which is absent for $SS < 0.2$ and $SS < 0.1$. However, we understand such estimation may roughly indicate the span of intersection in Silhouette space, and the exact position of the intersection between datasets cannot be inferred with confidence. Hence, we rectify ourselves as,

“We find a sharp change in the slope of $\langle EM \rangle$ profiles at cluster 3, considering $SS < 0.3$ and onwards, which is absent for the profiles corresponding to $SS < 0.2$ and $SS < 0.1$. Therefore, the span of intersection may roughly be estimated between 0.2 and 0.3 in the Silhouette space, and elements outside the intersection ($SS > 0.3$) are likely well-segregated among three blinking categories. At this point, although we refrain from concluding an exact extent of intersection, a sharp change in the slope of $\langle EM \rangle$ profiles can be an indicator providing scope to automate the decision in the future.” (Highlighted in yellow; page 5-6 of the Revised Manuscript)

Comment 6: In the SI, section 5, page 6: shouldn't B(k) be the inter- (not "intra") cluster dispersion?

Response: We thank the reviewer for bringing this mistake to our attention. We have now corrected the sentence as "The B(k) represents inter-cluster dispersion, which is the variance between cluster centroids." (Highlighted in yellow; page 6 of the Revised Supplementary Information)

Comment 7: The definition of "epoch"(= code iteration?) used in many figure labels in the SI is missing. (Remarks on code availability)

Response: Yes, "epochs" represents the number of code iterations. We have now mentioned it on page 5 of the Revised Manuscript as ".....performing 100 iterations (epochs) of the code...". We have slightly modified a sentence on page 17 of the Revised Manuscript as "To mitigate this discrepancy, we execute each code on the same dataset for 100 iterations (epochs) and consider the mean profiles of the Silhouette score (<SS>), CH index (<CH index>), and SSD values (<SSD>) for the trial clusters." (Highlighted in yellow; page 5 and page 17 of the Revised Manuscript)

Additional response (code availability): We have now provided below repository links for the code availability. These are mentioned under the section **Data availability, on page 18 of Revised Manuscript.**

1. <https://github.com/Amitrajit2024/Photoluminescence-Blinking-Data-Analysis.git> (for UML)
2. <https://zenodo.org/records/8027381> (for PSD analysis)

Reviewer #2 (Remarks to the Author)

General Comment 1: Debroye et al. present a manuscript describing an unsupervised machine learning algorithm to analyze the intensity fluctuations of nanocrystals recorded through widefield fluorescence microscopy. The algorithm and the mathematical tools utilized in the analysis are robust and there is no doubt that this methodology excels in the field and is of great intellectual value. It is, however, a manuscript focused solely on the algorithm, and I believe that for Nature Communications a broader application should be presented to be considered for publication.

Response: We sincerely thank the reviewer for finding "great intellectual value" of our methodology. We believe that multiple applications are crucial to consider a new methodology for publication in Nature Communications. For this purpose, we already demonstrated versatile applicability of our methodology addressing multiple challenges related to fluorescence microscopy of semiconductor nanocrystals as well as scanning tunnelling spectroscopy of the self-assemblies.

Three major challenges that we addressed are listed below,

1. Our methodology provides a generalized way to analyse heterogeneous photoluminescence blinking characteristics of an ensemble of semiconductor nanocrystals. This enables opportunity to characterize their optical quality through microscopy, in a time efficient manner, which would be particularly desired for industrial and academic research.
2. The same methodology demonstrates utility in detecting spatially synchronous blinking within a crystal assembly. This infers on the extent of charge carrier communication and (in)homogeneous blinking dynamics within the assembly (or thin film). This could be a metrology to investigate the extent of electronic synergy and diffusion length of the charge carriers within the system.
3. Our technique additionally proves utility in secluding the irrelevant current-voltage (I-V) signals obtained from scanning tunnelling spectroscopy, which originates from instrumental vibrations and drift of the tip location while operating at the solid-liquid interfaces at room temperature. This helps assigning I-V characteristics to the appropriate molecular components of an assembly, which is typically being conducted by manual interventions.

With automated solutions for photophysical and electrical characterization, we believe our methodology can also be explored in other scientific domains and therefore could be of interest toward the broad audience of Nature Communications. We have mentioned multiple usages as the strength of our methodology in the *Abstract*, last paragraph of the *Main*, and the *Discussion* sections (page 1, page 3 and page 14 of the Revised Manuscript, highlighted in green)

General Comment 2: The main application of the algorithm in the PL datasets is that it accurately categorizes them into three categories: a two-state, a PL enhancement and a PL Bleaching. Two-state blinking, PL enhancement (photobrightening) and PL bleaching (photodarkening) are processes well studied in the literature and are not new (J. Chem. Phys. 115, 1028–1040 (2001), Topics in Current Chemistry 2016, 374 (5), 58. Section 3.4). Furthermore, simpler algorithms have been established, not only to determine the behavior of the on-off trajectories, but for fluorescence lifetimes and spectral diffusion (J. Phys. Chem. A 2020, 124, 3494–3500, Phys Rev B 2014, 90 (19), 195302.) The authors should make an effort to provide -new- knowledge in the field of photophysics (or STM) while using the algorithm, only then this manuscript is worthwhile of Nature Communications.

Response: We agree with the reviewer that two-state blinking, PL enhancement (photo-brightening), and PL bleaching (photodarkening) are well-documented processes and that simpler algorithms exist for analysing ON/OFF dynamics. However, to the best of our knowledge, no existing methodology can simultaneously analyse all these blinking patterns across an ensemble of nanocrystals. Traditional two-state blinking analyses often face the challenge of setting subjective thresholds to distinguish ON/OFF states, and they cannot be applied to trajectories that include gradual photobleaching, photo-enhancement gradients, or complex multi-state blinking signatures.

What is new? Our contribution lies in providing a **universal and threshold-independent solution**. The unsupervised machine learning-power spectral density (UML-PSD) methodology efficiently segregates diverse blinking behaviours and delivers *category-specific information on intensity fluctuations and the nature of trap states*. Unlike previous approaches, this allows for ensemble-level analysis that incorporates not only two-state blinking but also trajectories with bleaching, brightening, or multiple emissive states.

Importantly, our approach enables **near-real-time analysis** (on the order of seconds to minutes, depending on dataset size and computational capacity). This makes it suitable for rapid evaluation of nanocrystal optical quality under various experimental conditions (e.g., surface passivation, excitation power, environment, temperature). We have discussed one application under comment 6. Beyond this immediate utility, the UML-PSD workflow lays the foundation for **smart microscopy platforms** capable of automated, high-throughput optical characterization of semiconductor nanomaterials. As we highlight in the revised Discussion (highlighted in green, on page 14 of the Revised Manuscript), such integration could accelerate both fundamental studies and industrial applications, especially as compact and chip-scale microscopes are emerging.

In summary, while the blinking processes are not new, our work advances the field by delivering the **first comprehensive, automated, and generalizable machine-learning framework** that simultaneously classifies heterogeneous PL blinking patterns, extracts insights on traps via PSD analysis, and points toward future “self-driving” microscopy solutions.

I believe that addressing the comments below could strengthen the impact of the manuscript.

Comment 1: A blinking process is the fingerprint of a single nanocrystal or molecule. How do the authors make sure that the multi-state-blinking (on and off) is the result of a single nanocrystal and not the of two or more NCs in the same diffraction limited spot? Although a second order correlation measurement (antibunching) is the definitive proof, the authors must provide clear evidence that the trajectories are indeed a single NC.

Response: We thank the reviewer for raising this important point. We fully agree that the blinking trajectory of a single nanocrystal (NC) is considered its fingerprint, and that antibunching measurements represent the definitive proof of single-emitter origin. However, the scope of the present study is to introduce an automated segregation-and-analysis framework that can handle large ensembles of trajectories in a time-efficient manner, rather than to replace labor-intensive, particle-by-particle confirmation experiments.

That said, we have strong reasons to believe that the dominant trajectories in our dataset indeed correspond to single nanocrystals. Photobleaching and photobrightening signatures, for instance, are highly unlikely to arise from multiple NCs within the same diffraction-limited spot, as it is improbable for two emitters to undergo bleaching or brightening with identical rates simultaneously. Likewise, two-state blinking is widely regarded as the fingerprint of a single NC or molecule. Based on our prior expertise in distinguishing PL fluctuations of single particles versus clusters (ACS Nano 2021, 15, 10775–10981; Adv. Mater. 2018, 30, 170549), we are confident that the clustering and PSD analysis reported here reflect single-NC dynamics for the major categories (bleaching, brightening, and two-state blinking).

For multi-state blinking traces, we agree that distinguishing whether they arise from one or multiple NCs in the same spot remains challenging. In line with the reviewer's concern, our present methodology treat such multi-state trajectories as potential outliers. We show that PSD analysis can already help identify these cases via usually high β values (see page 11 of the Revised Manuscript), and we envisage that supervised data-driven classification, trained on carefully validated datasets, will be a natural extension of this work.

Although we do not claim to provide definitive single-NC verification for every trajectory in the ensemble, our proposed methodology is scalable and reliably captures the mainstream single-NC blinking processes, while also offering pathways to detect, separate, and eventually study more complex multi-state behaviors.

In lights of the reviewer's comment, we have added three sentences (highlighted in green) on page 3-4 of the Revised Manuscript as "We consider photobleaching and photo-brightening trajectories to originate from single nanocrystals, as it is highly unlikely that multiple nanocrystals within the same diffraction-limited spot would undergo bleaching or brightening with identical rates. The same applies to two-state blinking, which is widely recognized as the fingerprint of a single nanocrystal or molecule. Based on our expertise in carrier dynamics and blinking analysis of NCs (and MCs)^{59,60} we are confident that our methodology predominantly reflects single-nanocrystal blinking phenomena."

Comment 2: Or, on this same train of thought, is the algorithm capable of categorizing the blinking trajectories and assigning them to two or more NCs in the same diffraction limited spot?

Response: This is a very interesting point raised by the reviewer. The clustering of PL intensities within a multi-state blinking trajectory (intra-trajectory clustering) is possible, which can provide information on the optimal number of intensity states therein. However, it will be difficult to predict an exact number of nanocrystals situated in the diffraction limited spot as the number of intensity states corresponding to each individual emitter is an unknown parameter.

Comment 3: While selecting the pixels from the .tiff file for the blinking trajectories, the authors claim that they were carefully selected after background subtraction with the use of ImageJ. Although it is not mentioned, I believe that this procedure is human-driven and may bias the subsequent analysis. I am wondering if the authors thought in applying a spatial analysis of the photons, similar to super resolution microscopy, to accurately and automatically assign the pixels that could correspond to a single NC. If the authors could bridge this new analysis to the one widely employed in super resolution microscopy

it would be of great value, not only for the photophysics and physical chemistry fields, but also for the biological sciences.

Response: We clarify that the extraction of blinking trajectories from the PL movies is mostly automated. After background flattening in ImageJ, a time-averaged frame is generated, from which only one manual step is introduced: cropping regions containing clusters or aggregates where multiple nanocrystals coexist. For the remaining diffraction-limited spots, ImageJ automatically identifies intensity maxima, and these (x,y) coordinates are used as the centre pixel of a 3×3-pixel area (diffraction limit) in MATLAB to extract intensity profiles. Thus, manual intervention is limited to removing clearly irrelevant regions, ensuring that the extracted signals correspond to individual emitters.

We completely agree with the reviewer that integrating super-resolution techniques, would further strengthen the methodology by enabling precise and automated localization of emitters. Such an extension could be particularly valuable for biological imaging, where sample complexity is greater. Our current focus has been on demonstrating the machine learning framework for PL blinking analysis; however, we see strong potential to bridge our workflow with super-resolution modules in future work. This integration would combine accurate spatial assignment with automated temporal analysis, thereby broadening the applicability of our method across photo-physics, physical chemistry, and biological sciences.

In response to the reviewer's comment, we added a sentence "Moreover, super-resolution techniques could be integrated upstream of the machine learning module to enable extraction and classification of time series with precise spatial localization. Such an approach would be particularly valuable for complex biological images, offering a pathway to extend our methodology beyond photo-physics of nanocrystals toward broader interdisciplinary applications." in the "Discussion" section (highlighted in green, page 14 of the Revised Manuscript). Also, we have now added "Data extraction" under the Methods section (highlighted in green, page 16 of the Revised Manuscript).

Comment 4: How does the analysis look like without normalizing the intensity trajectories, is there something that the algorithm can tell us about the heterogeneity in terms of brightness of the individual NCs?

Response: This is a good suggestion raised by the reviewer. The algorithm can perform intensity-based clustering on the un-normalized blinking traces. However, it will lack efficiency in producing pure clusters, due to misclassifications. This is because the PL intensities of trajectories can vary over a broad range and often are similar amongst different blinking patterns. Therefore, two unnormalized PL trajectories can be closer in Euclidean space even though their patterns are different. Hence, the algorithm will generate misclassification within the clusters. In light of the reviewer's comment, we have performed the intensity-wise clustering of the same dataset of blinking traces (photobleaching, two-state and photo-enhancement). The individual clusters are depicted below,

Review Fig. 1. Demonstrating K-means clustering of the unprocessed PL blinking trajectories considering three optimal clusters.

We find the misclassification is present in each cluster even after data-smoothing by applying binning window of 80 frames. The results demonstrate significance of normalization before clustering of the blinking patterns. The clustering of unnormalized binned trajectories is presented below,

Review Fig. 2. Representing K-means clustering of the unnormalized PL blinking trajectories after binning by 80 frames, considering three optimal clusters.

As PL intensities and fluctuation patterns are two independent features of blinking trajectories, we suggest clustering of one feature when the other is invariant (or almost similar). For this reason, we normalized all the PL intensity traces and brought them into the same intensity range before clustering their patterns. In other cases, where blinking characteristics are apparently similar throughout the space (microcrystal or film), we can segregate intensity zones by performing “intensity-wise clustering” without normalization. Then clusters will deliver information on spatially distributed “static quenchers”. We have demonstrated one such application under section 11 of the Supporting Notes (page 11 of the Revised Supplementary Information).

Comment 5: The time resolution (30ms) of the frame is defined by the end of the manuscript. The authors should use time and time-binning vocabulary throughout the manuscript and not frame-binning to make a closer connection to the photophysical processes well known in the literature.

Response: We thank the reviewer for this helpful suggestion. In line with the recommendation, we now clearly mention the time resolution of our data acquisition (30 ms) at the beginning of the Results section (highlighted in green on page 3 of the Revised Manuscript).

To align more closely with the photo-physics literature, we have also adopted the term “time-binning” throughout the manuscript wherever appropriate. At the same time, for clarity in data processing, we occasionally retain the term “frame-binning”, since each frame corresponds to one time stamp (30 ms) and thus defines the dimensionality of the dataset. **To avoid ambiguity, we have now mentioned “binned by W frames” is equivalent to “W×30 ms time-bins” on page 7, 12 of the Revised Manuscript and page 19 of the Revised Supplementary Information.** This allows the manuscript to remain consistent with well-established time-binning vocabulary while still highlighting the dimensionality aspect that is central to our analysis.

Comment 6: It is well known that different excitation fluences affect the photophysical behavior of the NCs (Nature 1996, 383 (6603), 802–804). The authors should provide analysis of the same sample, measured at different fluences, and demonstrate that the algorithm is capable of distinguishing and classifying the trajectories with significant differences.

Response: Blinking signature of nanocrystals can be affected by several parameters which we have discussed in the manuscript (**highlighted in green on page 2 of the Revised Manuscript**).

We agree with the reviewer that excitation fluence can strongly influence the blinking dynamics of nanocrystals, as elegantly demonstrated for CdSe QDs (Nature 1996, 383, 802–804). In principle, applying our methodology to datasets acquired at different fluences would indeed be a compelling validation. However, unlike CdSe QDs, perovskite nanocrystals degrade rapidly under increased excitation power in ambient conditions, which makes repeated measurements on the same sample at different fluences technically challenging and prone to artifacts. For this reason, we did not include fluence-dependent measurements in the present work.

Instead, to illustrate the capacity of our methodology to resolve changes in blinking behavior induced by external parameters, we present a complementary dataset comparing pristine and ligand-passivated (DDAB-treated) CsPbBr₃ nanocrystals. We have randomly chosen two regions of interest containing pristine (Review Fig. 3) and passivated (Review Fig. 4) nanocrystals and applied our methodology to cluster the blinking traces. For these regions, we produced the blinking statistics, where we studied 381 pristine and 128 DDAB-passivated CsPbBr₃ nanocrystals. We binned the PL trajectories by a window of 80 frames, followed by normalization, as proven beneficial for the data length of 10000 frames. Thereafter, we calculated the mean silhouette score (<SS>) for the processed ensembles and determined the optimal clusters. Next, we performed K-means clustering of the processed traces, which yielded the statistics. The results are described below,

Review Fig. 3. (a, b) The first frame and the last frame (b) of the PL movie of a representative area of interest containing pristine CsPbBr₃ nanocrystals. (c) The average silhouette score ($\langle SS \rangle$) for the ensemble of 381 emitters. (d) The unprocessed ensemble of the blinking traces corresponding to the pristine crystals. (e) Binned (by 80 frames) and normalized blinking trajectories. (f) The clusters of the processed blinking traces.

Pristine sample: We observe the pristine nanocrystals photobleached over time in ambient, which can be visualized by comparing the first and last frames of the representative PL movie (300 sec, at 30 ms time resolution), as shown in the Review Figure. 3a,b. Our methodology finds an elbow break at cluster 3 for the $\langle SS \rangle$ profile (Review Fig. 3c), indicating the optimal number of clusters. The unprocessed and the processed PL traces are shown in the Review Figure. 3d,e. The clusters (Review Fig. 3f) reveal the emitters underwent different extents of photo-bleaching. For instance, emitters in Cluster 2 (18%) demonstrate a slower photobleaching than in the Cluster 1 (48%) and Cluster 3 (34%). Simultaneously, the emitters in Cluster 3 likely exhibit comparatively fewer fluctuations compared to rest of the emitters.

Review Fig. 4. (a, b) The first frame and the last frame (b) of the PL movie of a representative area of interest containing DDAB-passivated CsPbBr₃ nanocrystals. (c) The average silhouette score ($\langle SS \rangle$) for the ensemble of 128 emitters. (d) The unprocessed ensemble of the blinking traces corresponding to the passivated crystals. (e) Binned (by 80 frames) and normalized blinking trajectories. (f) the clusters of the processed blinking trajectories.

Passivated sample: On the other side, we find several DDAB-passivated nanocrystals exhibit enhanced longevity till the last frame of photo-exposure, albeit some extent of photobleaching is present. The first and last frame of the representative PL movie of passivated crystals are compared in the Review Figure. 4a,b. The $\langle SS \rangle$ profile indicated the elbow break at the optimal cluster 3, as depicted in the Review Figure. 4c. Intriguingly, the $\langle SS \rangle$ value at the elbow is higher as compared to the pristine nanocrystals (Review Figure. 3c), indicating the blinking signatures are not necessarily photobleaching and heterogeneity raised in the ensemble of passivated crystals. The unprocessed and the processed PL traces are shown in the Review Figure. 4d,e. In the Review Fig. 4f, we find the emitters belonging to Cluster 1 (62.5%) exhibit photobleaching after a certain delay, and the delay time increases for the emitters in Cluster 3 (23.5%). Alongside, we find mostly two-state blinking and a few cases of photo-enhancement within the Cluster 2 (14%), and no photobleaching trend is evident.

Altogether, our unsupervised machine learning workflow successfully segregates sub-ensembles with distinct blinking statistics, clearly revealing the suppression of photobleaching and the emergence of more stable emission profiles upon surface passivation (Supplementary Note 10). The results provided here serve as a proof-of-principle that our algorithm can distinguish and classify trajectories with markedly different photophysical behaviours under varied sample conditions. We acknowledge that a systematic fluence-dependent study, ideally under controlled environments that mitigate perovskite degradation, would be a valuable extension and plan to pursue this in future work. In response to the reviewer's comment, we have now added these results under Supplementary Notes, section 10 and added a sentence on page 14 of the Revised Manuscript (highlighted in green).

Comment 7: Besides the “normal” blinking, the PL enhancement and PL bleaching, is the algorithm capable of determining photobleaching? i.e. QDs that were showing the on-off states and at certain time point it did not recover from the off state?

Response: To the best of our understanding, the reviewer perhaps indicated the following scenarios where a QD bleaches in a single step and never reappears, such as,

Review Fig. 5. Demonstration of the blinking behaviour of representative nanocrystals which undergo photobleaching in a single step and never come back to the emissive phase again.

The overall feature of these trajectories resembles “two-state” fluctuation due to a single-step bleaching process in absence of a continuous intensity gradient. Therefore, we expect these trajectories will be clustered as two-state blinking. However, it may be considered as a photobleaching category depending on “long duration” of the OFF/bleached state, as evident in the lower panel. Such specific cases could be classified in a better way by data training using supervised models.

Comment 8: Authors should provide more details of their detection scheme for PL measurements, that is, camera quantum efficiency, brand and model.

Response: The wide-field PL movies were collected using a Hamamatsu EM-CCD digital camera, model number C9100-13. The camera quantum efficiency is greater than 90% as per the model specifications. In light of the reviewer's comments, we have now provided these details in the Methods section. (highlighted in green, on page 16 of the Revised Manuscript)

Minor Comment 1: Fig 1. The text font inside the figure is too small. There was no code in ESI.

Response: According to the reviewer's comment, we have now modified Fig. 1 in the Revised Manuscript with an enhanced font size of the text.

Minor Comment 2: There was no code in ESI.

Response: We have now provided below repository links for the code availability. These are mentioned under the section **Data availability, on page 18 of Revised Manuscript**.

1. <https://github.com/Amitrajit2024/Photoluminescence-Blinking-Data-Analysis.git> (for UML)
2. <https://zenodo.org/records/8027381> (for PSD analysis)

Reviewer #3 (Remarks to the Author)

General Comments: This work proposes the use of machine learning algorithms for the optical characterization of the “quality” of all-inorganic perovskite NCs via their emitting blinking behavior. At least for II-VI semiconductor NCs the blinking behavior together with the profile of the excited state (PL lifetime) have been used to assess the “quality of these NCs, a good NC exhibiting a two-state on-off blinking with a single exponential lifetime. Perovskites NCs are a little more challenging in the blinking behavior with existing reports showing blinking even from micron sized crystals (see the works of Scheblikyn in NanoLetters 2015), and this makes the proposed assessment challenging, at least when it comes to judging the “good” optical quality of a perovskite compound, either NC or microcrystal. Because of this, I commend the authors for the grand challenge they have taken. I also think that the use of power spectral density (PSD) in calculating the blinking behavior is preferred against other methods, in particular when dealing with such heterogeneous NCs, see the work of Pelton (APL 2004) who demonstrated the use of PSD to uncover blinking even in ensembles of emitters in solution and not necessarily isolated NCs. So the proposed method of Hofkens and team is interesting as it extends to ensemble of emitters, which is done here to a certain extend by correlating synchronous blinking from multiple grains in a crystal. Having the ability to classify charge carrier dynamics with spatio-temporal accuracy is important to the community and to be able to classify traps in perovskites is important as well, and for large samples this involves large data sets where ML can help. Also the synchronous blinking among traps in a single crystal is a nice demonstration of the capability of this ML method. All together I find the study interesting and innovative.

Response: We sincerely thank the reviewer for understanding the innovation and finding our methodology interesting. We believe that the “segregation and analysis” workflow will serve both the academic and industrial research on large-scale nanocrystals. Particularly, it will give an opportunity to investigate blinking trajectories beyond typical two-state characteristics.

Comment 1: The authors could comment how their method could be expanded to solution systems where blinking is also present although spatial info may be lost.

Response: We appreciate the reviewer's suggestion and discussed the possibilities below.

It is likely that the dispersion of nanocrystals will generate an ensemble-averaged PL fluctuation where sharp “blinks” will be suppressed due to the averaging of signals. This can lead to loss of information, and therefore inferences on individual QDs will be muted. Pelton *et al.* (in APL 2004) reported 1/f characteristics of power spectral density for an ensemble of CdSe QDs in solution and compared it with that for the individual immobilized QDs, which revealed a close similarity. However, the blinking spectra for QD solution was not available (corresponding to Figure 1 of Pelton *et al.*, APL 2004) which perhaps could provide a better understanding on temporal PL fluctuations.

Moreover, in past years, “two-state” blinking dynamics of individual immobilized QDs is observed heterogeneous and non-ergodic (J. Phys. Chem. C 125, 2021, 17133–17143). Hence, it may not always be true that PSD information for a dispersion of QDs will represent the blinking behaviour of individual QDs, and it may bias the interpretation. It is also impossible to deconvolute the ensemble PL fluctuation of a dispersion into arbitrary number of components representing contribution of single QDs. Furthermore, blinking of perovskite nanocrystals often deviates from two-state pattern and undergo photobleaching, brightening etc. Therefore, the ensemble PL fluctuation of perovskite nanocrystals (in solution) will be severely influenced by statistical weightage of differently blinking sub-populations and will not reflect the individual scenario. Hence, we are confident that PL imaging of segregated and immobilized nanocrystals on the coverslips, and application of UML-PSD methodology for class-wise analyses of blinking traces, offers a better opportunity to understand nonradiative charge-carrier dynamics in an ensemble of nanocrystals.

In response to the reviewer’s comment, here we discuss a plausible method which could briefly infer on the ensemble quality of QDs in solution, given no diffusion of particles within the focal volume and no laser oscillations contributing to the overall PL fluctuation. Under this circumstance, we first need to set a reference “good” QD solution which will provide the standard PL fluctuation trajectory and the corresponding PSD profile. The goodness refers to apparently uniform blinking (or suppressed blinking) characteristic of individual QDs, which can be investigated under the microscope. Thereafter, several batch-wise QD solutions of unknown quality will be investigated maintaining identical experimental parameters, and their PSD profiles will be generated. Now, these PSD profiles can be clustered using UML technique and the clusters can be assigned as “good” or “bad” samples by comparing their closeness to the reference one.

Additional response (code availability): We have now provided below repository links for the code availability. These are mentioned under the section **Data availability, on page 18 of Revised Manuscript**.

1. <https://github.com/Amitrajit2024/Photoluminescence-Blinking-Data-Analysis.git> (for UML)
2. <https://zenodo.org/records/8027381> (for PSD analysis)

Detailed point-by-point author response to referee's comments

We are grateful to the editor and all the reviewers for the constructive feedback and acceptance of our manuscript, "*Machine Learning for Microscopy Data Analytics Targeting Real-time Optical Characterization of Semiconductor Nanocrystals*" as an article in Nature Communication. We have now modified our manuscript according to the editorial requests following the CHECKLIST provided to us. At the same time, we respectfully acknowledge the final concern raised by the *Reviewer 2* and have incorporated some lines in the main manuscript as well as added two figures in the Supplementary Information. Below we provide a comprehensive response to the concern raised.

Reviewer #2 (Remarks to the Author)

General Comments: Debroye et al. present a revised manuscript that effectively addresses the reviewers' concerns and demonstrates significant improvement. I commend the inclusion of analyses on unnormalized data sets and nanocrystals with varied passivation ligands. However, I find a contradiction in the authors' rebuttal letter that warrants clarification.

The authors' response addressing the general comment 2 states: "Our contribution lies in providing a universal and threshold-independent solution. The unsupervised machine learning-power spectral density (UML-PSD) methodology efficiently segregates diverse blinking behaviours and delivers category-specific information on intensity fluctuations and the nature of trap states," Yet, in addressing comment 4, they acknowledge: "The algorithm can perform intensity-based clustering on the unnormalized blinking traces. However, it will lack efficiency in producing pure clusters, due to misclassifications...".

This discrepancy should be addressed. The authors ought to acknowledge in the manuscript that the algorithm's performance is limited when applied to less processed data. Specifically, it is unclear how the method can be considered threshold-independent if it struggles with unnormalized traces. Clarifying this point would strengthen the manuscript's claims and improve transparency.

Once this minor revision is incorporated, I believe the paper is suitable for publication in Nature Communications, and no further revisions are necessary.

Response: We sincerely thank the reviewer for acknowledging the improvements in our revised manuscript and for providing constructive suggestions that help further strengthen the work.

In response to the reviewer's concern regarding the apparent discrepancy, we would like to clarify that the primary objective of our methodology is the segregation of blinking patterns through unsupervised clustering. For this purpose, we compute Euclidean distances between blinking profiles, which necessitates normalization of blinking traces at the initial stage to bring all the patterns within the same data range (0 to 1). Normalization is therefore an essential step before performing shape-dependent clustering of blinking profiles, to ensure that shape-dependent features, rather than absolute intensity values, drive the clustering outcome. Furthermore, 'normalization and clustering' methodology is completely different from the threshold-dependent calculations for two-state blinking analysis. For this reason, normalization does not represent a limitation of the UML-PSD methodology, nor does it contradict our claim of threshold-independence, which refers specifically to the avoidance of manually defined intensity thresholds used in two-state blinking analysis.

We acknowledge that in certain cases the spatial blinking dynamics may be homogeneous across a microcrystal (or thin film) while the absolute intensity range varies. This may originate from spatially synchronous dynamics of transient traps in the presence of spatially varied density of static nonradiative quenchers. In such scenarios, intensity-based (i.e., unnormalized) clustering becomes relevant to probe

spatial distributions of static quenchers. We already demonstrate such an application in Section 11 of the Supporting Notes (page 12 of the Supplementary Information). Thus, whether normalized or unnormalized data will be analysed depends on the specific scientific question being addressed.

To avoid any confusion, we have now added a few explanatory texts to explicitly clarify this point (highlighted in yellow on pages 4, 6, 8 of the main manuscript), and included two supplementary figures (Supplementary Fig 16 and 17) illustrating the clustering results for unnormalized-raw and unnormalized-processed blinking traces.

The Zenodo repository links (DOIs) for code and data availability are now provided on page 18 and 19 in the revised manuscript, and also mentioned in the references 81 and 82. The repository links are <https://doi.org/10.5281/zenodo.17715957> (for UML analysis) and <https://zenodo.org/records/8027381> (for PSD analysis).

We thank the reviewer again for helping us improving the reasoning and transparency of our manuscript.